# Active site recovery and N-N bond breakage during hydrazine oxidation boosting the electrochemical hydrogen production

Libo Zhu[1,2,6], Jian Huang [1,6], Ge Meng[1,2], Tiantian Wu[1], Chang Chen[1,2], Han Tian[1], Yafeng Chen[3], Fantao Kong[1], Ziwei Chang[4], Xiangzhi Cui [1,2,5] ✉ & Jianlin Shi [1,2] ✉

Substituting hydrazine oxidation reaction for oxygen evolution reaction can result in greatly reduced energy consumption for hydrogen production, however, the mechanism and the electrochemical utilization rate of hydrazine oxidation reaction remain ambiguous. Herein, a bimetallic and hetero-structured phosphide catalyst has been fabricated to catalyze both hydrazine oxidation and hydrogen evolution reactions, and a new reaction path of nitrogen-nitrogen single bond breakage has been proposed and confirmed in hydrazine oxidation reaction. The high electro-catalytic performance is attributed to the instantaneous recovery of metal phosphide active site by hydrazine and the lowered energy barrier, which enable the constructed electrolyzer using bimetallic phosphide catalyst at both sides to reach $500\ mA\ cm^{-2}$ for hydrogen production at 0.498 V, and offer an enhanced hydrazine electrochemical utilization rate of 93%. Such an electrolyzer can be powered by a bimetallic phosphide anode-equipped direct hydrazine fuel cell, achieving self-powered hydrogen production at a rate of $19.6\ mol\ h^{-1}\ m^{-2}$.

Electro overall water splitting (OWS) is an efficient approach to obtaining high-purity hydrogen by using clean energy such as wind and solar energy[1]. However, the anodic oxygen evolution reaction (OER) in OWS is kinetically slow, which necessitates high electrolytic potential and large amount of expensive catalyst to overcome the gigantic reaction energy barrier[2]. Thus, alternative strategies have been developed to reduce the energy input for OWS. Coupling the electrocatalytic oxidation of the small molecules with low reaction energy barrier, such as methanol[3], ethanol[4], formic acid[5], hydrazine ($N_2H_4$) and urea[6], etc. in the anodic chamber is one of the effective strategies to reduce the energy consumption for hydrogen evolution reaction (HER)[7,8], which could concurrently electro-synthesize the high valued products.

Among various small molecule oxidation reactions, hydrazine oxidation reaction (HzOR, $N_2H_4+4OH^-\rightarrow N_2+4H_2O+4e^-$, −0.33 V vs. RHE) with absolute carbon-free products and less catalyst poisoning is an ideal energy-saving substitute for OER (1.23 V vs. RHE)[9,10]. Unfortunately, however, $N_2H_4$ tends to decompose directly to form $N_2$ and $H_2$ ($N_2H_4\rightarrow N_2+2H_2$), which would reduce the electrochemical utilization rate of $N_2H_4$[11]. Thus, designing efficient and low-cost HzOR electrocatalysts to enhance the electrochemical utilization rate of $N_2H_4$ is of great significance to achieve low energy consumption and high efficiency for hydrogen production.

Recently, the HzOR electrocatalysts have been investigated extensively including the Pt based noble metal[12] and transition metal

[1]State Key Lab of High Performance Ceramics and Superfine Microstructure, Shanghai Institute of Ceramics, Chinese Academy of Sciences, Shanghai 200050, P. R. China. [2]Center of Materials Science and Optoelectronics Engineering, University of Chinese Academy of Sciences, Beijing 100049, P. R. China. [3]Collaborative Innovation Center of Steel Technology, University of Science and Technology Beijing, Beijing 100083, P. R. China. [4]School of Physical Science and Technology, Shanghai Tech University, Shanghai 201210, P. R. China. [5]School of Chemistry and Materials Science, Hangzhou Institute for Advanced Study, University of Chinese Academy of Sciences, Hangzhou 310024, P. R. China. [6]These authors contributed equally: Libo Zhu, Jian Huang. ✉e-mail: cuixz@mail.sic.ac.cn; jlshi@mail.sic.ac.cn

based catalysts[13,14]. Among these electrocatalysts, transition metal phosphide has been reported to be the efficient catalyst toward HzOR because of its advantages of high chemical stability and outstanding conductivity[15–17]. However, the HzOR/HER performances under high current density are not satisfactory due to the limited gas-solid-liquid three-phase reaction interface at the catalyst. Most recently, three-dimensional (3D) nanoarrays structure has been reported to be feasible for enhancing the HzOR/HER catalytic performances because of its highly exposed active sites, good hydrophilicity as well as the fast charge/mass transfer ability[18,19]. Though the HzOR/HER performances are elevated at a certain degree[20], the catalytic mechanism probing of the most reports still focuses on HER, while the HzOR of 4-electron transfer process, which is important to enhance the electrochemical utilization rate of $N_2H_4$ in the anode chamber, has been little known, leading to the ambiguous HzOR active centers and indeterminate reaction path of HzOR[21,22].

Herein, a 3D nickel cobalt phosphide heterostructure loaded on nickel foam (Ni-Co-P/NF) with CoP nanoparticles being uniformly distributed on the NiCoP nanowires was fabricated by an interface engineering strategy and used as a probe catalyst for the first time to reveal the HzOR mechanism and elevate the electrochemical utilization rate of $N_2H_4$ as well. The heterostructure nanoarrays and the hydrophilic/hydrophobic interface structure endow the Ni-Co-P/NF bifunctional catalyst with much enhanced activities toward both HER and HzOR, especially, only 176 mV is needed to reach a high current density of 1000 mA cm$^{-2}$ for HzOR. The superior electrochemical activity of Ni-Co-P/NF heterostructure nanoarrays originates from the interface electron transfer between CoP nanoparticles and the NiCoP nanowires, resulting in the extremely low energy barrier of HzOR and even a new reaction path of nitrogen-nitrogen single (N-N) bond breakage at 0.2 V or above. Additionally, the in situ formed MPO$_x$ (metal phosphorus oxide) resulting from the electrochemical oxidation of Ni-Co-P/NF in anodic chamber can be instantly recovered to active MP (metal phosphide) species during HzOR, ensuring the 100 h durability at 100 mA cm$^{-2}$. Resultantly, the voltage input of the electrolyzer equipped with Ni-Co-P/NF catalysts for both HER and HzOR is as low as 0.88 V to reach 200 mA cm$^{-2}$, 1.04 V lower than that of OWS (1.92 V), greatly reducing the energy consumption. Especially, the $N_2H_4$ electrochemical utilization rate reaches 93% by optimizing the $N_2H_4$ concentration. Furthermore, the electrolyzer can be powered by the self-assembled direct hydrazine fuel cell (DHzFC) using Ni-Co-P/NF as anodic catalyst for HzOR, realizing self-powered hydrogen production at a rate up to 19.6 mol h$^{-1}$ m$^{-2}$ without external electricity supply.

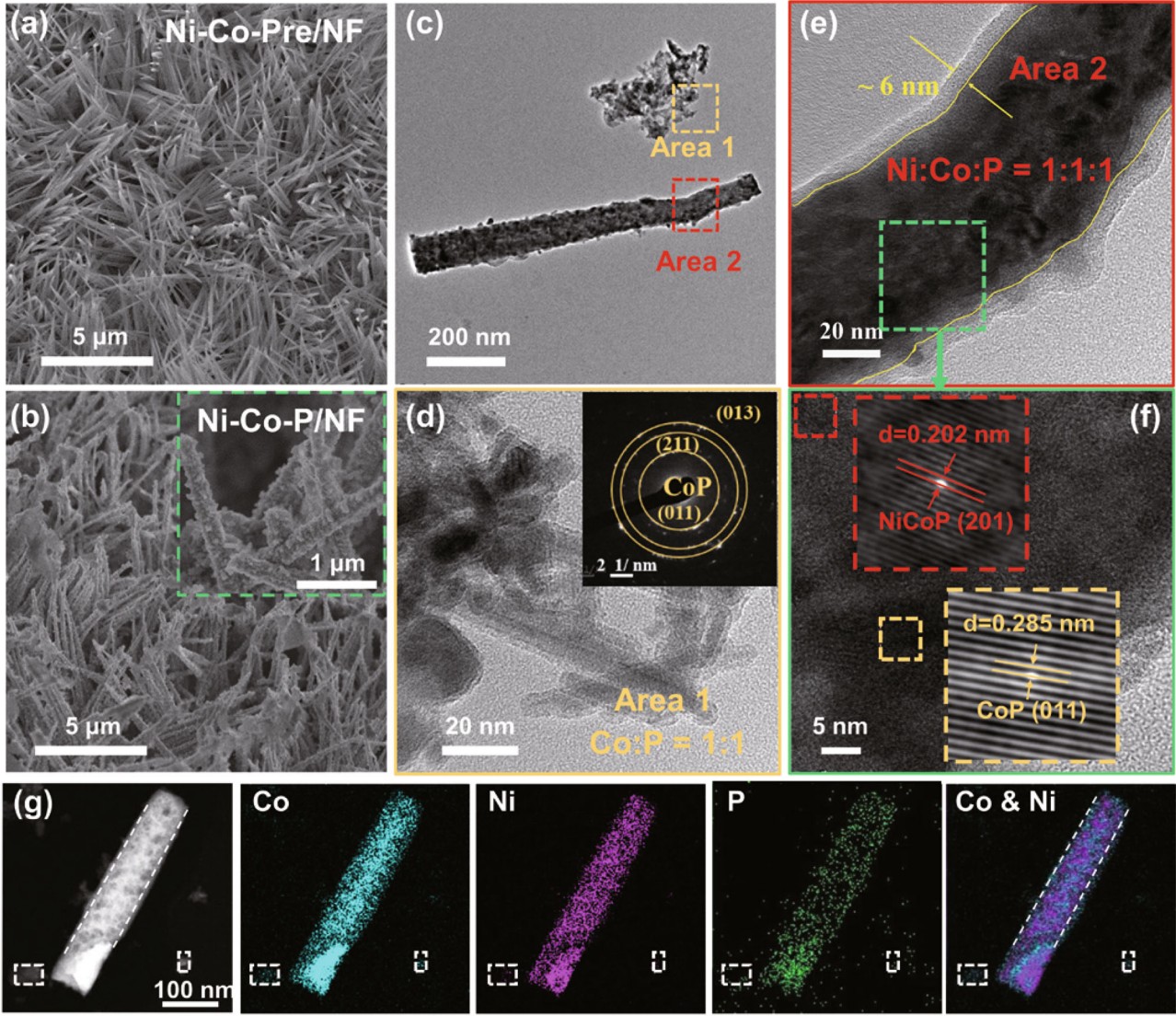

**Fig. 1 | Morphological characterizations of Ni-Co-P/NF and reference samples.** SEM images of **a** Ni-Co-Pre/NF and **b** Ni-Co-P/NF with a magnified image in the inset; **c** TEM image of Ni-Co-P/NF, in which the squared Area 1 and Area 2 are magnified and shown respectively in (**d**) and (**e**) with a FFT pattern of the Area 1 in the inset of (**d**); **f** HRTEM image of the Ni-Co-P/NF with a further magnified image in the inset; **g** STEM-EDS elemental mapping of Ni-Co-P/NF.

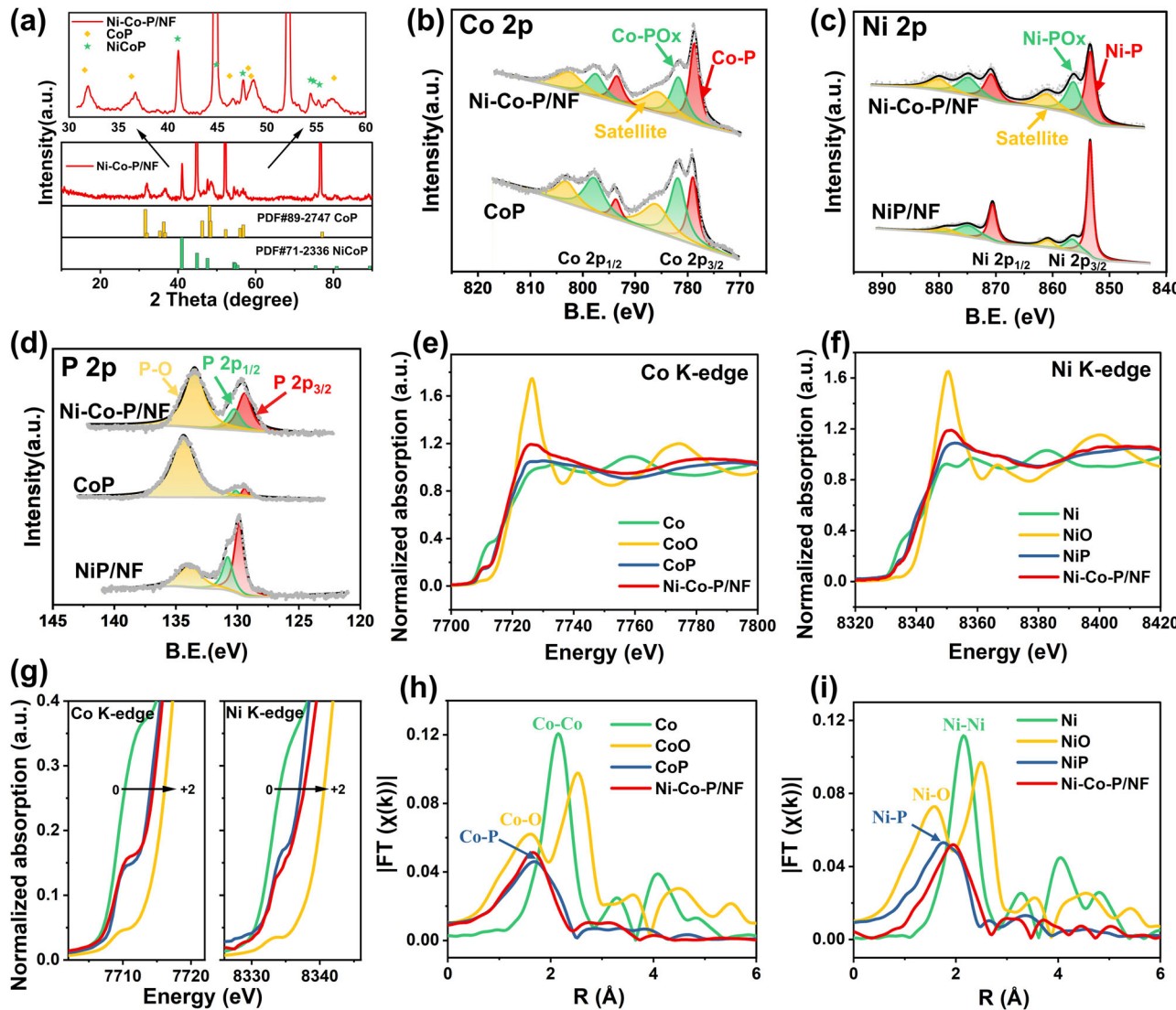

**Fig. 2 | Structural characterizations of Ni-Co-P/NF and reference samples. a** XRD patterns of Ni-Co-P/NF; High-resolution XPS spectra of **b** Co 2p; **c** Ni 2p; **d** P 2p for the prepared samples; **e** Co and **f** Ni K-edge XANES spectra of Ni-Co-P/NF and the reference compounds; **g** the left and right correspond to the squared areas in (**e**) and (**f**), respectively; EXAFS results of (**h**) Co K-edge; (**i**) Ni K-edge for Ni-Co-P/NF and the reference compounds.

## Results and discussion

### Catalyst characterization

The Ni-Co-P heterostructure nanoarrays with CoP nanoparticles uniformly distributed on the surface of NiCoP nanowires were in situ synthesized on nickel foam (NF) by ion exchange method as shown in Fig. S1, in which the NF is corroded and oxidized to Ni (II) during the hydrothermal synthesis to form Ni-Co precursor nanoarrays on NF (Ni-Co-Pre/NF), and then the Ni-Co-P/NF was synthesized after phosphating by using NaH$_2$PO$_2$[23,24].

The Ni-Co-Pre/NF shows a nanowire array morphology of 900 nm in an average length on NF (Fig. 1a, Fig. S2). With the increase of Co source addition amount from 3 to 5 mmol, the nanowire arrays of Ni-Co-Pre/NF gradually accumulate into a sea urchin shape (Fig. S3a, b), which adhere on the NF substrate loosely and will easily fall off in subsequent treatment. 3 mmol cobalt source addition amount is optimal, at which the precursor loading on NF achieves maximum according to the XRD patterns and the digital photographs (Fig. S3c). After phosphating, sample Ni-Co-P/NF at the optimal Co source addition of 3 mmol retains the dense and uniform 3D nanowire morphology of precursor (Fig. 1b, Fig. S4), while the surface becomes rougher because of a large number of nanoparticles formed on the surface of

nanowires (inset in Fig. 1b). Moreover, the nanoarrays of Ni-Co-P/NF are hydrophilic featuring a much lower water contact angle (36.14°) than that of Pt/C (139.98°) (Fig. S5), which is favorable for the release of absorbed bubbles on discontinuous solid-liquid-gas triple phase contact points[25].

In the TEM and high-resolution TEM (HRTEM) images (Figs. 1c, d), two different morphologies in Ni-Co-P (the sample separated from NF) were observed: randomly oriented particles from nanowires by ultra-sonic treatment (area 1) (Fig. 1d) with the selected area electron diffraction (SAED) pattern being assignable to CoP with (011) (211) (013) planes (Fig. 1d illustration), and nanowire main phases (area 2) (Fig. 1e). The energy dispersion spectrum (EDS) (Fig. S6) indicates the component ratio of area 1 is Co:P = 1:1 and that of area 2 is Ni:Co:P = 1:1:1, further confirming the existence of CoP in area 1 and nickel cobalt phosphide (NiCoP) in area 2. While the edge area of the nanowire (Fig. 1e) is Co-rich and Ni-deficient besides P signal according to the element linear scanning profile in Fig. S7, evidencing the existence of CoP nanoparticles outside of the NiCoP nanowires. In addition, the oxygen signal can also be detected, indicating the partial oxidation of the outer layer into an amorphous phosphorus oxide layer[26]. Two sets of interplanar spacing of 0.202 nm and 0.285 nm can be detected in

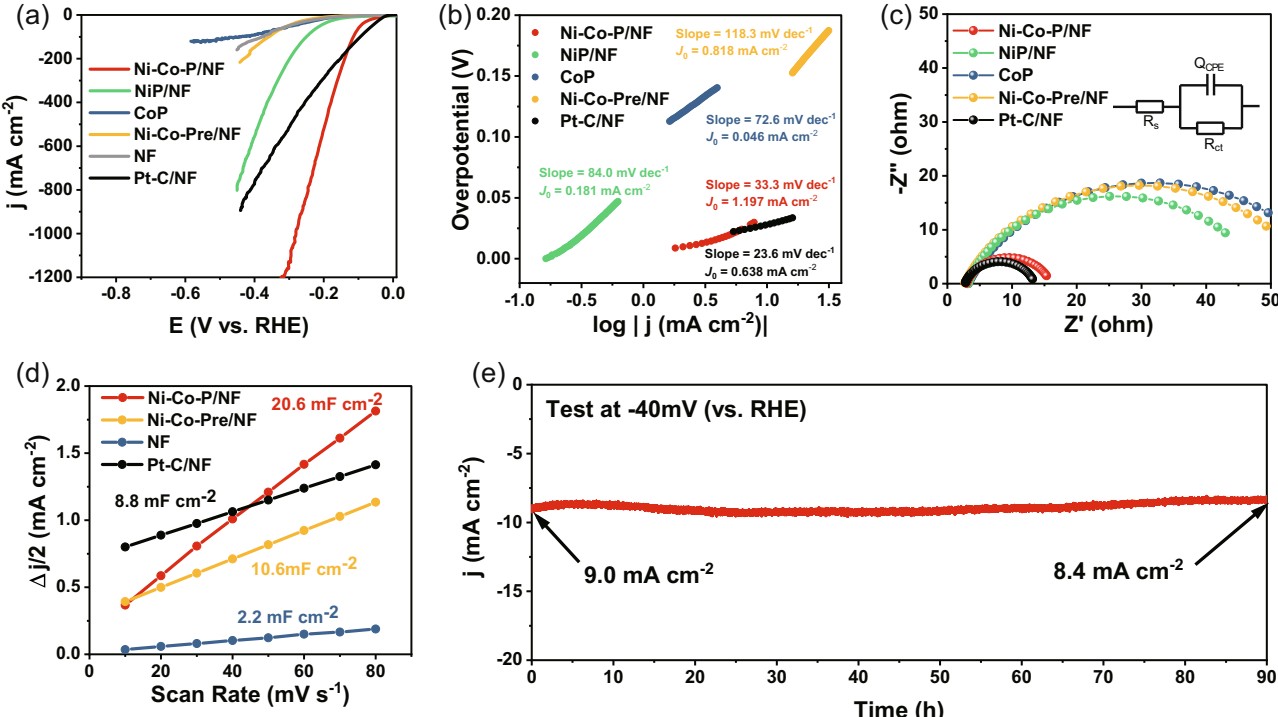

**Fig. 3 | Electrochemical properties of HER in N$_2$-saturated 1.0 M KOH. a** 30% iR-corrected LSV curves at scan rate of 5 mV s$^{-1}$ (The compensation resistance is about 0.6 Ω with the corresponding non-iR compensated data shown in Fig. S10b); **b** Tafel plots and exchange current densities; **c** Nyquist plots with the fitting pattern in the inset; **d** Double-layer capacitance C$_{dl}$; **e** Stability measurement of Ni-Co-P/NF. The mass loadings of Ni-Co-P/NF, Ni-Co-Pre/NF, NiP/NF and Pt-C/NF are about 4.42, 4.80, 2.27 and 2.00 mg cm$^{-2}$, respectively.

HRTEM (Fig. 1f), which are in correspondence to the (201) plane of NiCoP and (011) plane of CoP, respectively, indicating the successful synthesis of Ni-Co-P heterostructure nanoarrays with CoP nanoparticles uniformly distributed outside of the NiCoP nanowires in Ni-Co-P/NF (Fig. S8). The element mappings in Fig. 1g demonstrate the homogeneous dispersion of Co, Ni, P on nanowires, while Co is slightly wider than that of Ni, further indicating successful loading of CoP nanoparticles on the NiCoP nanowires surface.

From the XRD patterns in Fig. 2a, the strongest diffraction peaks of Ni-Co-P/NF centered at 40.99°, 44.90° (covered by nickel peaks), and 47.58° can be indexed to the (111), (201), (210) planes of standard hexagonal NiCoP (JCPDS No. 71-2336), and those at 31.59°, 36.32°, 48.12° assigned to the (011), (111), (211) planes of CoP (JCPDS No. 29-0497), further verifying the formation of Ni-Co-P heterostructure nanoarrays on NF with CoP nanoparticles distributed outside of the NiCoP nanowires[27].

The survey scan of XPS spectrum of the Ni-Co-P/NF indicates the existence of Co, Ni, P, and O element (Fig. S9). Co 2p XPS spectra (Fig. 2b) of Ni-Co-P/NF exhibits two major groups with binding energies of 778.7 and 793.5 eV, respectively, representing Co-P, which are close to those of metallic Co (778.2 and 793.3 eV), indicating the presence of partially charged Co species (Co$^{δ+}$, δ is close to 0). While the relatively weak bands at 781.8 and 797.6 eV belong to the Co-PO$_x$ resulting from the Co oxidation state, and the 785.6 and 802.5 eV are the satellite peak of Co. Similarly, there are three groups of peaks appear in the Ni 2p XPS spectra (Fig. 2c), and the binding energies at 853.6 and 870.8 eV represent Ni-P (Ni$^{δ+}$), and those at 56.8 and 874.7 eV belong to Ni-PO$x$, and the 861.3 and 879.6 eV are assigned to the satellite peak of Ni[28–31]. Compared to CoP, the Co 2p$_{3/2}$ and 2p$_{1/2}$ of Ni-Co-P/NF slightly shift toward the lower binding energies (Fig. 2b, c, Fig. S9), while the Ni 2p$_{3/2}$ and 2p$_{1/2}$ of Ni-Co-P/NF have a higher binding energy shift compared NiP/NF, indicating an increased ionicity of M − P bond in bimetallic phosphides and a promoted electron migration

from metal to phosphide[32], which makes the electronic structure of the Ni-Co-P more conducive to the HER and HzOR. The deconvoluted P XPS peaks (Fig. 2d) at 128.3 and 130.2 eV belong to P 2p$_{1/2}$ and P 2p$_{3/2}$ of MP, which is lower than that of elemental P (130.0 eV), suggesting that the P is partially negatively charged (P$^δ$), and the peak at 133.5 eV could be ascribed to a metallic oxidation state, which is associated with MPO$_x$ due to exposed to the air[33]. In addition, the binding energies of P in MP and MPO$_x$ in Ni-Co-P/NF also shift to the lower position compared with those of NiP/NF and CoP, indicating the electron-rich character of P. Thus, the P can trap positively charged protons during electrocatalysis, which is responsible for the HER activity[34].

X-ray absorption fine structure (XAFS) spectrum was obtained to investigate the local electronic structure and atomic arrangement of Co K-edge and Ni K-edge. In the normalized X-ray absorption near edge structure (XANES) (Fig. 2e), the absorption edge energy of Co K-edge in Ni-Co-P/NF is in between those of CoO and Co foil (Fig.2g-left), and similar to that of CoP, indicating that the average value state of Co element is in between 0 and +2, consistent with the XPS results[30]. Similarly, the absorption edge energy of Ni K-edge of Ni-Co-P/NF gives the average value states of Ni between 0 and +2 (Fig. 2f), slightly higher than that of NiP according to Fig. 2g-right, suggesting the changed binding energy between Ni and P due to the electronic interaction between Co and Ni. The Fourier-transformed extended X-ray absorption fine structure (EXAFS) spectrum of Ni-Co-P/NF shows the similar radial distribution function to those of CoP and NiP as shown in Fig. 2h and Fig. 2i, respectively. Relative to the reference CoP, a similar peak at 1.66 Å in Ni-Co-P/NF is assigned to Co−P bond because of the existence of CoP species in Ni-Co-P/NF (Fig. 2h). Notably, the Ni−P peak in Ni-Co-P/NF positively shifts to 1.75 compared to 1.69 Å in NiP reference (Fig. 2i), due to the Co introduction. All above results demonstrate the interaction between Ni and Co atoms in NiCoP lattice and the coexistence of CoP and NiCoP components.

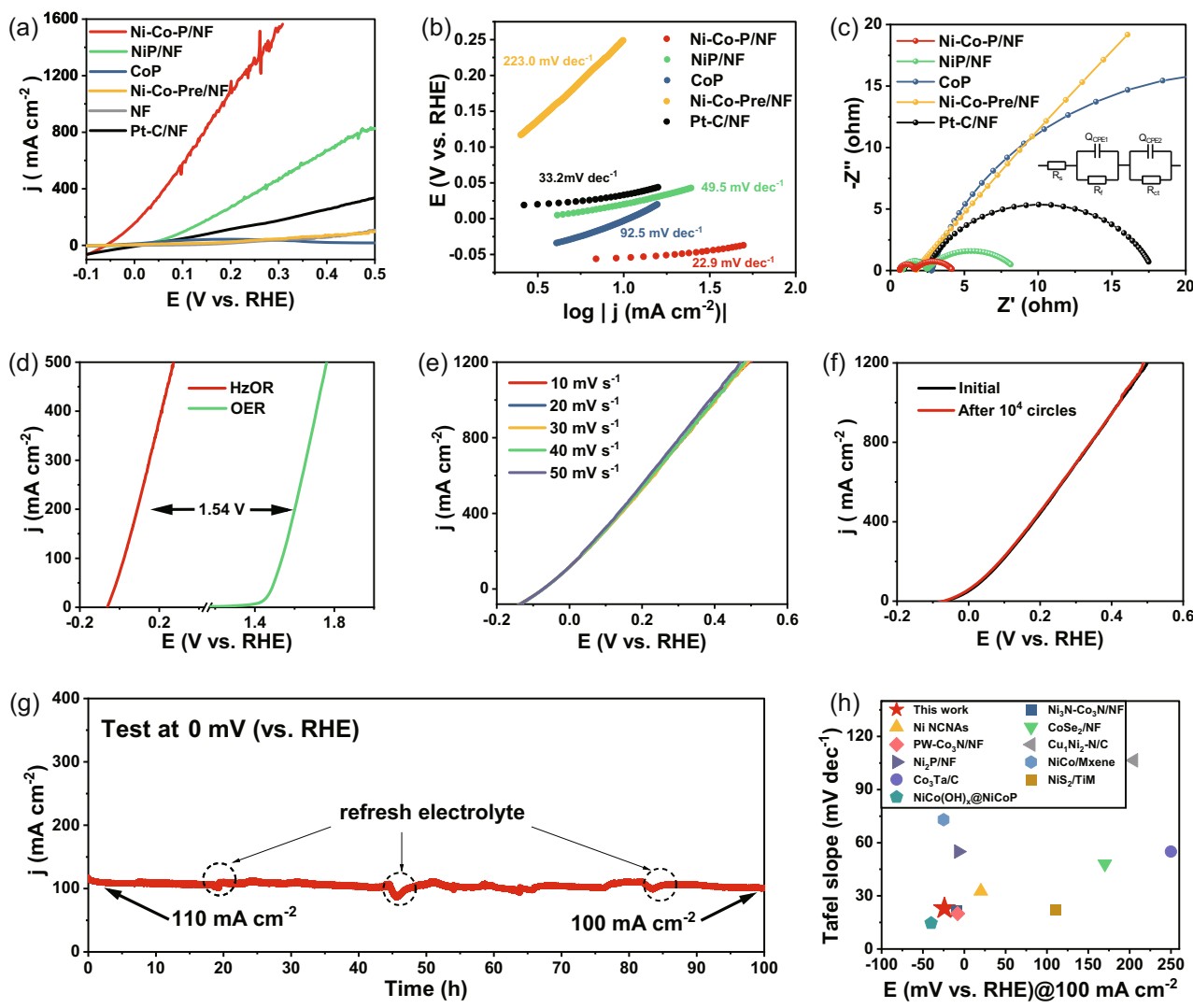

**Fig. 4 | Electrochemical properties of the HzOR in 1.0 M KOH and 0.1 M N₂H₄.** **a** 50% iR-corrected LSV (The compensation resistance is about 0.31 Ω with the corresponding non-iR compensated data shown in Fig. S16a); **b** Tafel slope plots; **c** Nyquist plots with the fitting pattern in the inset; **d** Polarization comparison between HzOR of Ni-Co-P/NF and OER of RuO₂; **e** LSV curves of Ni-Co-P/NF varying with scanning speed; **f** LSV curves of Ni-Co-P/NF before and after ADTs; **g** Stability measurements of Ni-Co-P/NF for HzOR; **h** Comparison of the potentials at 100 mA cm⁻² and Tafel slopes between Ni-Co-P/NF in this work and various non-noble metal-based catalysts reported.

## Electrocatalytic HER performance

The HER electrocatalytic activity of the as-synthesized catalysts was investigated in $N_2$-saturated 1.0 M KOH electrolyte, which was calibrated by reversible hydrogen electrode (RHE) as shown in Fig. S10a. From the linear sweep voltammetry (LSV) curves in Fig. 3a (the non-iR compensated data in Fig. S10b), sample Ni-Co-P/NF requires a much lower overpotential (37 mV) to reach 10 mA cm⁻² compared to references NiP/NF (124 mV), Ni-Co-Pre/NF (175 mV), and CoP (173 mV). Though the overpotential of Ni-Co-P/NF below 150 mA cm⁻² is slightly higher than that of the Pt-C/NF ($\eta_{10}$ = 28 mV), but become substantially lower than that of Pt-C/NF at current density above 150 mA cm⁻². Importantly, the Ni-Co-P/NF catalyst needs only 280 mV to reach 1000 mA cm⁻², which is better than most reported catalysts (Table S1). A Tafel slope of 33.3 mV dec⁻¹, much smaller than those of NiP/NF (84.0 mV dec⁻¹), Ni-Co-Pre/NF (129.3 mV dec⁻¹), and CoP (72.6 mV dec⁻¹), though slightly larger than 23.6 mV dec⁻¹ of Pt-C/NF in the potential range, was recorded on Ni-Co-P/NF (Fig. 3b)[35], suggesting the Tafel step as rate determination step for HER and the rapid increase of HER rate with increase of overpotential on Ni-Co-P/NF. The exchange current density ($J_0$) from the intercept of the linear region in Fig. 3b for Ni-Co-P/

NF is 1.197 mA cm⁻², about twice of that of Pt-C/NF (0.638 mA cm⁻²), meaning that the catalyst Ni-Co-P/NF can supply large currents even at very low overpotential and the HER can be easily activated with fast electrode kinetics. Typically, the $J_0$ is thought to be proportional to catalytically active surface area, manifesting that the high catalytic activity of Ni-Co-P/NF comes mainly from the large active area by interface engineering construction.

The electrochemical resistance spectra (EIS) of the catalysts are shown in Fig. 3c, and the resistance of Ni-Co-P/NF (13.09 Ω) is much lower than those of CoP (57.3 Ω) and NiP/NF (45.9 Ω) though slightly higher than that of the Pt-C/NF (10.73 Ω) according to the fitting data in Table S2. Since the electrochemical double-layer capacitance ($C_{dl}$) of the catalyst is proportional to its electrochemical active surface area (ECSA) directly related to catalytic activity, Ni-Co-P/NF shows the $C_{dl}$ value of 20.6 mF cm⁻² (Fig. 3d and Fig. S11), much higher than Ni-Co-Pre/NF (10.6 mF cm⁻²), NF (2.2 mF cm⁻²), and Pt-C/NF (2.2 mF cm⁻²), consistent with the results of $J_0$, indicating the highest ECSA of Ni-Co-P/NF heterostructure nanoarrays among the materials prepared.

In addition, the Ni-Co-P/NF shows superior HER catalytic stability, which can be operated stably for 90 h at −40 mV with a limited current

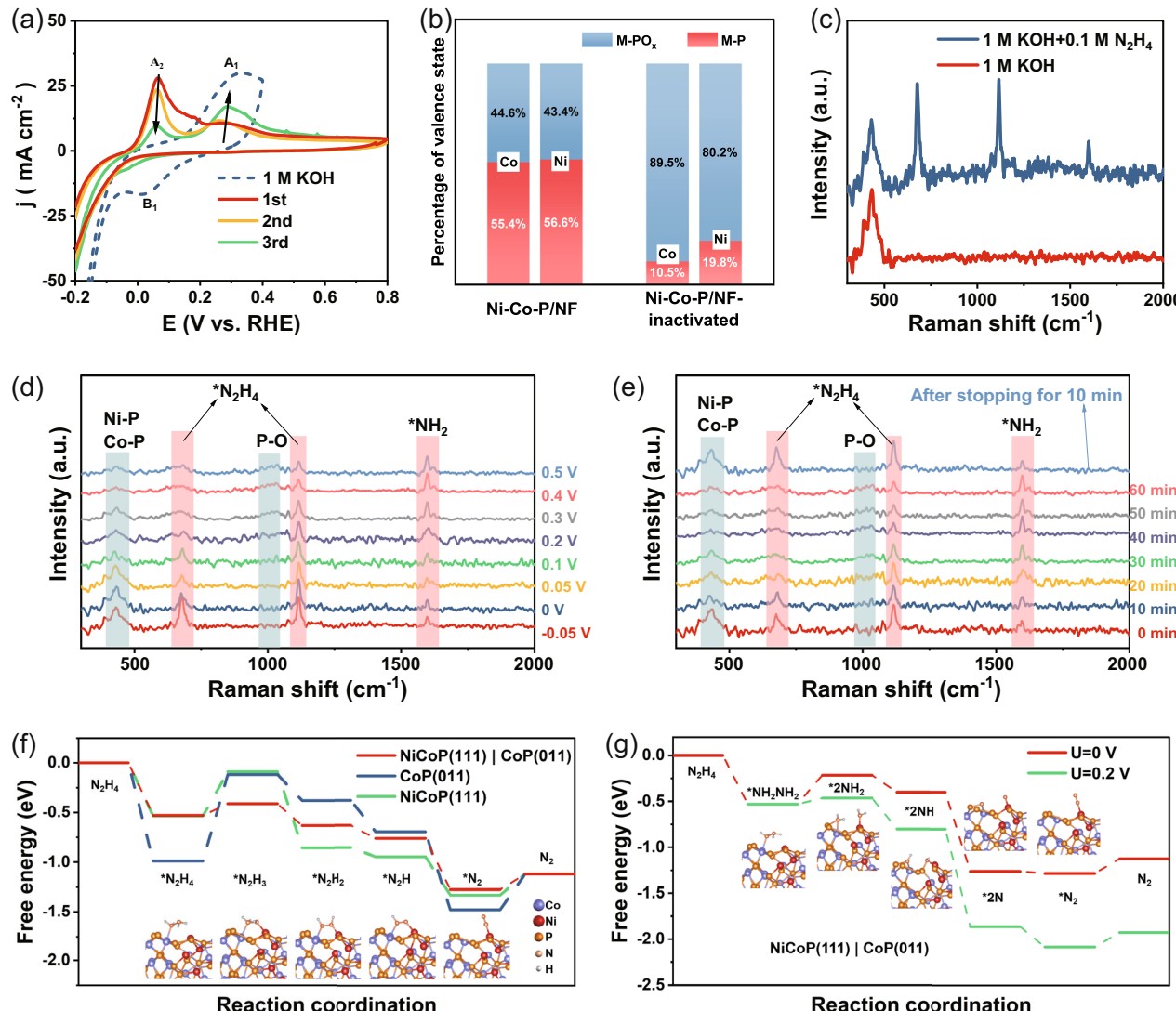

**Fig. 5 | Catalytic mechanism of HzOR and the catalyst's evolution during catalysis. a** CV curves in 1 M KOH (blue curves) or 1.0 M KOH + 0.005 M $N_2H_4$ under different cycles of scanning (red: first cycle, yellow: second cycle and green: third cycle); **b** Histograms of the ratio between $MPO_x$ and MP for the surface of Ni-Co-P/NF and Ni-Co-P/NF-inactivated; **c** Raman spectra of Ni-Co-P/NF in 1 M KOH or 1 M KOH + 0.1 M $N_2H_4$; (**d, e**) In situ electrochemical Raman spectra of Ni-Co-P/NF in 1 M KOH + 0.1 M $N_2H_4$ (**d**) at varied applied potentials and (**e**) at different reaction time intervals under constant 0.2 V; (**f**) Free energy changes of HzOR in path I at NiCoP(111)/CoP(011) interface, CoP(011) and NiCoP(111) surfaces and the most stable adsorption configuration of the intermediate at NiCoP(111)/CoP(011) interface in the inset; (**g**) Free energy changes of HzOR in path II at NiCoP(111)/CoP(011) interface with the most stable adsorption configuration of the intermediate.

attenuation by 6.7% at the end of test (Fig. 3e). This catalyst even can work durably for hydrogen production at as low as −160 mV exporting a current density of about 100 mA cm⁻² (Fig. S12). The accelerated durability tests (ADTs) in Fig. S13 show no significant change in LSV curves after 10000 cycles, further confirming the outstanding stability of Ni-Co-P/NF toward HER. Furthermore, the influence of Co addition on the HER performance of Ni-Co-P/NF composite was also tested (Fig. S14), and the Co-3 with Co(NO₃)₂·6H₂O addition of 3 mmol during the synthesis (as named Ni-Co-P/NF above) is optimal for HER catalytic activity. Moreover, the volume of H₂ production was collected by gas chromatography (GC) and compared with the theoretical gas volume obtained from the number of transferred electrons, which gives a HER Faradaic efficiency (FE) of 97.0% on Ni-Co-P/NF (Fig. S15).

## Electrocatalytic HzOR performance
The HzOR electrocatalytic activity of the as-synthesized catalysts was tested in an aqueous electrolyte containing 1.0 M KOH and 0.1 M N₂H₄ using a typical three-electrode system. Figure 4a exhibits the LSV curves

with 50% iR compensation at the scan rate of 50 mV s⁻¹ (the non-iR compensated data in Fig. S16a). Compared to NiP/NF (20 mV), CoP (−3 mV) and Ni-Co-Pre/NF (260 mV), sample Ni-Co-P/NF requires a much lower potential (−61 mV) to reach a current density of 10 mA cm⁻², which is superior to the Pt-C/NF (34 mV). Exceptionally, the low working potentials of −24, 78 and 176 mV are required for Ni-Co-P/NF heterostructure nanoarrays to reach current densities of 100, 500 and 1000 mA cm⁻², respectively. From Fig. 4b, Ni-Co-P/NF shows a small Tafel slope of 22.9 mV dec⁻¹, largely lower than those of Pt-C/NF (33.2 mV dec⁻¹), NiP/NF (49.5 mV dec⁻¹) and CoP (92.5 mV dec⁻¹), indicating its fast catalytic kinetics for HzOR. The superior HzOR catalytic activity of Ni-Co-P/NF is connected with its much smaller electrochemical resistance. In the simulated equivalent circuit diagram in the inset of Fig. 4c, $R_s$, $R_f$ and $R_{ct}$ respectively represent the solution resistance, high-frequency semicircle resistance and the charge-transfer resistance, and the $Q_{CPE1}$ and $Q_{CPE2}$ were introduced to simulate the double-layered capacitor at catalyst interface and the electrode/electrolyte interface, respectively[36]. According to the result of EIS and

model fitting (Table S3), Ni-Co-P/NF shows the lowest $R_{ct}$ (2.614 Ω) and $R_f$ (0.990 Ω) values owing to its three-dimensional array structure having greatly increased contact area at gas-solid-liquid three-phase, in which the $N_2H_4$ molecules adsorbed on the catalyst surface and electrolyte can be oxidized immediately and continuously, meanwhile the electrons generated can be efficiently transferred through conductive NF.

Figure 4d gives the comparison between HzOR by Ni-Co-P/NF (in 1.0 M KOH and 0.1 M $N_2H_4$) and OER by $RuO_2$ catalyst (in 1.0 M KOH). A potential of 90 mV is required for Ni-Co-P/NF to reach an elevated current density of 200 mA cm$^{-2}$, which is 1.54 V lower than that of $RuO_2$ catalyst (1.63 V) at the same current density, suggesting the significantly reduced energy consumption for hydrogen production by replacing OER with HzOR. To further investigate the HzOR kinetics for Ni-Co-P/NF, varied scanning rates were adopted. It can be seen from Fig. 4e that the current changes in LSV curves are negligible at the varied scanning rates from 10 to 50 mV s$^{-1}$, suggesting the efficient charge and mass transfer at the electrode-electrolyte-gas three-phase interface on Ni-Co-P/NF, and resultantly, the fast HzOR kinetics. The Ni-Co-P/NF with Co source addition of 3 mmol shows the better HzOR performance (Fig. S16b), coincident with the better HER activity. Moreover, the LSV curve of Ni-Co-P/NF after 10000 cycles shows no significant shifts compared to the initial cycle according to the ADTs (Fig. 4f), indicating its superior catalytic stability. From the i-t test of Ni-Co-P/NF toward HzOR (Fig. 4g), a large current density above 100 mAcm$^{-2}$ can be retained, which is 91% of the initial value at the end of 100 h. In addition, the average FE of HzOR at different potentials with Ni-Co-P/NF in 1.0 M KOH + 0.1 M $N_2H_4$ is shown in Fig. S17, and rather high FEs of HzOR can be obtained in all voltage ranges with the highest FE of 97 % achieved at 0.1 V vs. RHE. Furthermore, the phase structure, composition and morphology of the Ni-Co-P/NF catalyst after stability test (name as Ni-Co-P/NF-used) remain almost unchanged after the test according to the XRD patterns (Fig. S18), Raman spectra (Fig. S19)[37] and TEM images (Fig. S20), further confirming the HzOR catalytic stability of Ni-Co-P/NF heterostructure nanoarrays. The HzOR activity parameters of Ni-Co-P/NF, such as the potential to reach 100 mAcm$^{-2}$ and Tafel slope in this study, are even better than those of the most reported non-noble metal-based catalysts (Fig. 4h, Table S4).

## HzOR mechanistic insight in high activity and stability

The superior stability of bimetallic phosphide for HzOR has been investigated as follows. The CV measurements on Ni-Co-P/NF electrode were firstly carried out in the voltage range of −0.2 V to 0.4 V in 1 M KOH and −0.2 V to 0.8 V in 1.0 M KOH + 0.1 M $N_2H_4$ to determine the active sites of bimetallic phosphides and the potential changes on HzOR (Fig. 5a). A pair of redox peaks, $A_1$ and $B_1$ on the dotted blue CV curve, corresponds to the conversion between $MPO_x$ (the valence state of M is in between +2 and +3) and MP (M: $\delta^+$, $\delta$ is close to 0) in the 1 M KOH, indicating the surface oxidation/reduction of the metal phosphide in the anodic oxidation range as shown in Eq. (1).

$$MP + OH^- \leftrightarrow MPO_x + H_2O + e^- \qquad (1)$$

When adding 0.005 M $N_2H_4$ in electrolyte, a new oxidation peak $A_2$ appears in the CV curves, which is related to the preferential oxidation of $N_2H_4$ to the MP oxidation. The area of oxidation peak $A_2$ decreases with the increasing cycling number due to the decreased $N_2H_4$ concentration, while that of oxidation peak $A_1$ increases gradually because of the consumption of local $N_2H_4$ resulting in the following oxidation of the catalyst itself into $MPO_x$. In particular, the reduction peak ($B_1$) disappears in $N_2H_4$-containing electrolyte even under the existence of oxidation peak $A_1$, indicating that no $MPO_x$ species is present on the catalyst surface, or can be spontaneously reduced back to MP, if there is any, by $N_2H_4$ because of its strong reductivity as

shown in Eq. (2). The result suggests that the in situ electrochemically oxidized Ni-Co-P/NF in anodic potential range can be recovered back to active MP species again by $N_2H_4$.

$$MPO_x + N_2H_4 \rightarrow MP + N_2 + H_2O \qquad (2)$$

However, after the long-time CV measurements of Ni-Co-P/NF under a lowered $N_2H_4$ concentration or without $N_2H_4$, the HzOR activity of Ni-Co-P/NF will decline rapidly and cannot be recovered even at increased $N_2H_4$ concentration in the following (Fig. S21), which is due to the over-oxidation of catalyst (named Ni-Co-P/NF-inactivated) leading to the deactivation of Ni-Co-P/NF in the absence of $N_2H_4$. From the XPS spectra of Ni-Co-P/NF-inactivated (Fig. 5b, Fig. S20c, d), it can be found that the ratio of $MPO_x$ to MP significantly increased on the surface of catalysts, proving the over-oxidation of Ni-Co-P/NF responsible for the deteriorated HzOR performance. In addition, the EDS results of Ni-Co-P/NF and Ni-Co-P/NF-inactivated samples show that the percentage of oxygen in the catalyst is increased significantly after deactivation (Fig. S23), which is in accordance with the conversion from MP to $MPO_x$ (Fig. 5a), further confirming that the MP is the active sites for HzOR. These active sites can be in situ recovered by $N_2H_4$ molecules in the electrolyte in the case of being transiently oxidized to achieve long-lasting durability.

In order to further verify the mechanism, in situ Raman measurements were carried out to explore the compositional and structural changes of the catalyst. In 1 M KOH electrolyte, the catalyst Ni-Co-P/NF shows a significantly broadened band at around 400 cm$^{-1}$ corresponding to the vibration of M−P bond[38]. From the in situ Raman spectra (Fig. S24) shown at varied applied voltages in 1 M KOH, a new peak (corresponding to the stretching bond of P−O) appears at 1020 cm$^{-1}$ when the voltage is about 0.2 V, and its intensity increases with the elevation of voltage, while that of M-P decreases correspondingly, verifying the transformation from MP to $MPO_x$ under voltage in Eq. (1)[39,40]. When adding 0.1 M $N_2H_4$ in electrolyte, three strong $N_2H_4$ related peaks appear at 677, 1116, 1598 cm$^{-1}$ (Fig. 5c), respectively assigned to N−H, N−N stretching modes of adsorbed *$NH_2NH_2$, and N−H bending modes of intermediate *$NH_2$ on the Ni-Co-P/NF surface[41–43]. Figure 5d shows the real-time Raman spectra at varied applied voltages for monitoring the catalytic process by Ni-Co-P/NF for HzOR in 1 M KOH + 0.1 M $N_2H_4$ (−0.05 V to 0.5 V vs. RHE). At the voltage higher than 0.1 V, the peak density of M−P (400 cm$^{-1}$) gradually becomes weakened with the increase of voltage, and a weak peak appears near 1020 cm$^{-1}$ assignable to the stretching bond of P-O in $MPO_x$, indicating that part of MP species is gradually oxidized to $MPO_x$ during the HzOR. From the in situ Raman measurements at varied reaction time intervals at constant 0.2 V (Fig. 5e), when the voltage application was stopped for 10 min, the peak intensity of M−P bond (400 cm$^{-1}$) increased again, while the peak of P−O bond (broad band at about 1020 cm$^{-1}$) disappeared, confirming that the slightly oxidized species ($MPO_x$) could be reduced and recovered back to active MP species due to the presence of $N_2H_4$, which agrees well with the CV measurement in Fig.5a. Based on the above experiments, it can be concluded that the bimetallic phosphide, which is easily oxidized to eliminate activity, can be continuously reconstructed by $N_2H_4$ to expose its active sites for the good durability during HzOR.

More interestingly, when the applied voltage is greater than 0.2 V in Fig. 5d, the peak of *$NH_2$ becomes significantly stronger indicating the accumulation of reaction intermediate *$NH_2$. Different from the traditional reaction path (path I), the $NH_2$ group comes from N-N bond breakage of $N_2H_4$, suggesting another possible path for HzOR at voltages higher than 0.2 V. Firstly the $N_2H_4$ molecule firstly is adsorbed on the Ni-Co-P/NF surface to form *$NH_2NH_2$, in which the N−N bond will break to form two *$NH_2$ groups. Each *$NH_2$ then gradually dehydrogenates to generate $N_2$. The two paths in HzOR are in accordance

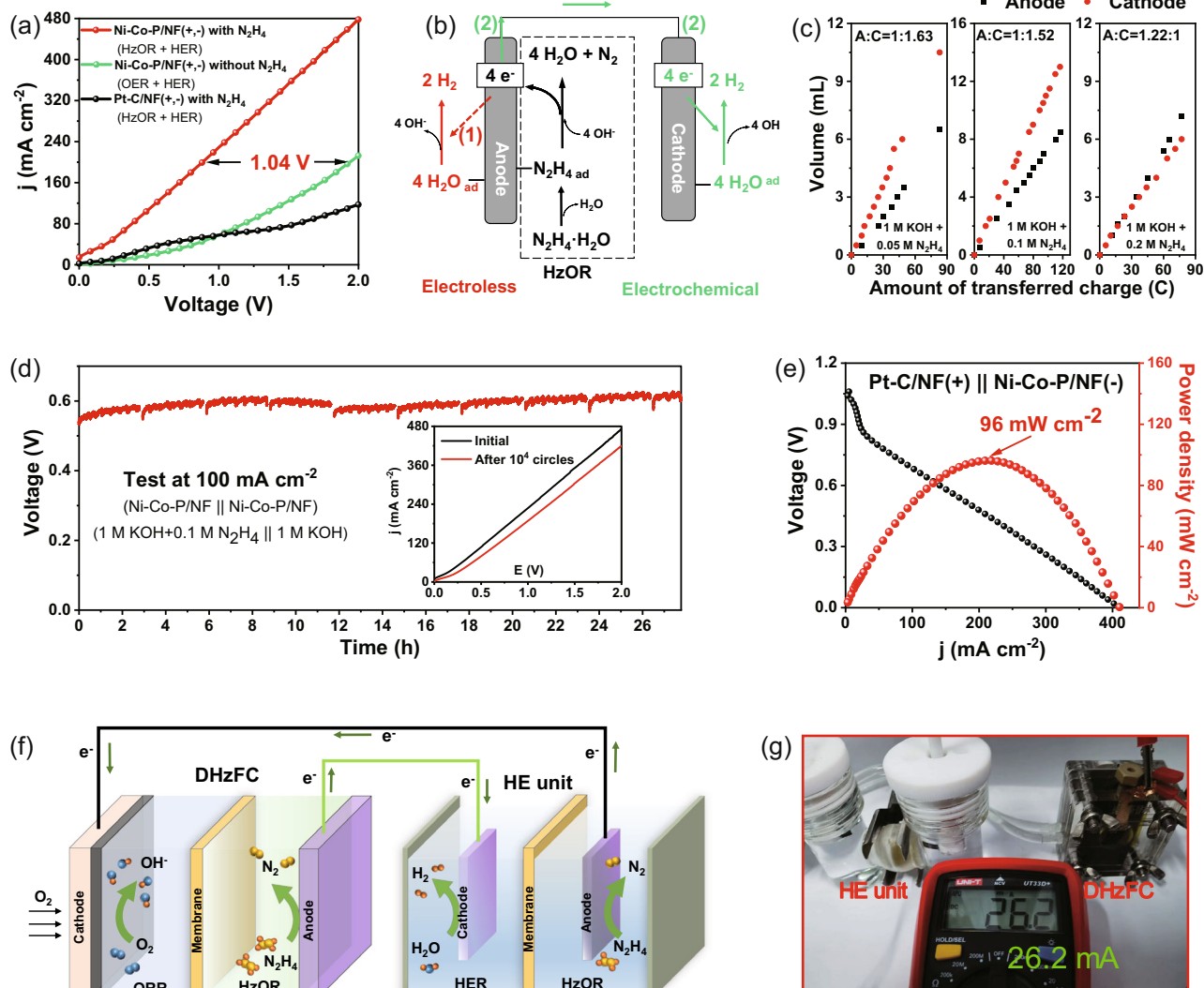

**Fig. 6 | Electrochemical properties of the HE unit (a, b, c, d) with Ni-Co-P/NF as both anode and cathode catalysts; DHzFC (e) with Ni-Co-P/NF as anode and Pt-C/NF as cathode; Self-powered System (f, g). a** LSV curves of the electrolyzer equipped with different electrode catalysts in 1 M KOH without or with the addition of $N_2H_4$ at anode (without iR compensation); **b** The proposed competing electroless decomposition path (red, path 1) and electrochemical path for $H_2$ production from hydrazine (green, path 2); **c** Relationship between gas volume and transferred charge number at anode and cathode under different anodic $N_2H_4$ concentrations; **d** Stability measurements at 100 mA cm$^{-2}$ with the ADTs in the inset; **e** Discharge polarization curve and power density plots; **f** Schematic illustration of a HE unit self-powered by a home-made DHzFC for $H_2$ production; **g** Digital photograph of $H_2$ production system with HE unit self-powered by DHzFC.

with the reverse processes of the dissociative and associative mechanisms in nitrogen reduction reaction[44].

In order to prove the promotion of heterojunction to HzOR and the possible N-N bond breakage mechanism, density functional theory (DFT) calculations were performed. As shown in Fig. S25, the NiCoP(111)/CoP(011) interface was constructed to model the NiCoP-CoP heterostructure of the Ni-Co-P/NF catalyst according to the HRTEM analysis (Fig. S8). When in contact with electrolyte, the as-established model is stable, and trace amounts of Co, Ni elements are dissolved from the catalyst surface into the electrolyte in the measured HzOR (Fig. S26). The density of states (DOS) of the NiCoP-CoP (Fig. S27) shows clearly the metallic nature, and the $d$-band center is located at −1.432 eV (spin-up) and −1.416 eV (spin-down), which are close to the values of other heterostructures enabling superior bifunctional activity[45]. From the Bader charge analysis (Fig. S28), it can be found that more charges from Co can transfer to P than from Ni, showing that the $N_2H_4$ will be preferentially adsorbed in the Co site on the heterostructure, followed by the Ni site. Based on this, five different kinds of adsorption sites at the heterointerface have been identified and

screened, and $\Delta G_{*N_2H_4}$ at the Co site (−0.530 eV) is optimal (Fig. S29). Therefore, the most favorable adsorption site of $N_2H_4$ is Co atom terminals, consistent with the above conclusion that MP is the active site. For comparison, the theoretical model on NiCoP (111) and CoP (011) sites are also illustrated (Fig. S30). The oxidation of $N_2H_4$ on the catalyst surface is usually a four-continuous-step proton coupled electron transfer (PCET) process--path I[46,47]. The free-energy of adsorbed intermediates and dehydrogenation steps at the NiCoP(111)/CoP(011) interface, and on NiCoP(111) and CoP(011) surfaces were calculated and shown in Fig. 5f. The rate-determining step (RDS) on NiCoP-CoP is *$N_2H_4$ conversion to *$N_2H_3$, and the $\Delta G$ value of this step is as low as 0.118 eV, much lower than that on CoP(011) surface (0.872 eV) and NiCoP(111) surface(0.444 eV), indicating that the NiCoP-CoP heterointerface can offer a fast kinetics for HzOR owing to the effectively modulated electronic structure.

Furthermore, according to the in situ Raman spectroscopic result, another theoretically possible reaction path, path II featuring N−N bond breakage, was proposed and probed. The charge density difference analysis shows that the N atom in *$N_2H_4$ has a significant charge

transfer to the nearby Co and Ni atoms, leading to the N−N bond length increase and the resultant activation of $*N_2H_4$ molecules (Fig. S31)[48]. As shown in Fig. 5g, at U = 0 V, the N−N bond breakage ($\Delta G =$ 0.315 eV) to form $2*NH_2$ is the RDS, while HzOR is more thermodynamically feasible to take place through path I than path II. However, with the increase of the potential to 0.2 V (U = 0.2 V), the N atoms can be more strongly bound onto the catalyst surface, leading to the much lowered $\Delta G$ value down to 0.065 eV, implying the greatly weakened N−N bonding. Therefore in this case, HzOR follows a new reaction path (path II) of N-N bond breaking in hydrazine molecules in addition to the traditional four-continuous-step PCET path.

Moreover, the mutual promotion relationship between metal phosphide and hydrazine may inspire the future researches for similar reactions, such as metal phosphide used for sodium borohydride ($NaBH_4$) oxidation reaction (BOR), or transition metal sulfide, selenide and other catalysts for HzOR. Similar to HzOR, $NaBH_4$ can also re-activate the oxidation-deactivated MP in BOR by recovering active centers of partially oxidized MP again (Fig. S34a and S34b), endowing the catalyst with good BOR durability (Fig. S34c).

## HE Unit Powered by DHzFC

Considering the electrocatalytic performance of the Ni-Co-P/NF for both HER and HzOR, a H-type double-chamber electrolyzer (HE unit) was assembled using the Ni-Co-P/NF as both cathodic and anodic catalysts. The HE unit takes 1.0 M KOH + 0.1 M $N_2H_4$ and 1 M KOH as the anodic and cathodic electrolytes, respectively, separated by anion exchange membrane (AEM). The separation design can avoid the pollution of $N_2H_4$ to the cathode solution to obtain high-purity $H_2$ at cathode. As shown in Fig. 6a, a current density of 200 mA cm$^{-2}$ can be obtained at the cell voltage of 0.88 V when using Ni-Co-P/NF as electrodes for HE unit, while it can only reach 50 mA cm$^{-2}$ under the same voltage by using Pt-C/NF electrodes (Fig. 6a). In sharp contrast, the traditional OER + HER requires 1.92 V to achieve 200 mA cm$^{-2}$, which is 1.04 V higher than that with Ni-Co-P/NF electrodes (0.88 V) under the same conditions, indicating the greatly reduced energy consumption by using HzOR assistant hydrogen production.

When no voltage is applied, the HE unit still exhibits an open circuit voltage (OCV) of −0.1 V and current output (Fig. S35), indicating the spontaneous $H_2$ evolution in the device. As a matter of fact, both HzOR and HER will take place concurrently on the Ni-Co-P/NF catalyst surface in anode chamber (Fig. 6b, path 1), leading to the electroless "spontaneous decomposition" of $N_2H_4$ without current (electron) output towards cathode, which is actually not desirable for anode $N_2H_4$ oxidation-coupled cathode HER, but widely ignored and indeed inevitable[49]. In order to further study this phenomenon, the gases produced at the cathode and anode were collected by drainage collection method to analyze the actual utilization rates ($\eta$) of $N_2H_4$ (Fig. S36). The potential for measurements was kept in a very low range (<0.2 V) to avoid the activation of anodic OER by hydrazine[50], meaning that $N_2$ is the only oxidation product on anode and the utilization rate of $N_2H_4$ can be calculated indirectly according to the gas volume of the anode. The electroless "spontaneous decomposition" of $N_2H_4$ can be inhibited by controlling the $N_2H_4$ concentration as shown in Fig. 6c, so as to promote the electrochemical oxidation rate of $N_2H_4$ (Fig. 6b path 2). The "spontaneous decomposition" can be largely prevented at the $N_2H_4$ concentrations lower than 0.05 M, and in this case the electrochemical utilization rate of $N_2H_4$ (in path 2 in Fig. 6b) reaches 93.0%. Comparatively, the "spontaneous decomposition" reaction will take place vigorously at increased $N_2H_4$ concentrations, and the utilization rates of $N_2H_4$ were calculated 90.5% and 67.6% at 0.1 M and 0.2 M $N_2H_4$, respectively.

The HE unit exhibits good stability for $H_2$ evolution (Fig. 6d), and the current density of 100 mA cm$^{-2}$ can be driven at the voltage lower than 0.63 V and operated stably for nearly 30 h. During the long-term hydrogen production (Fig. 6d) with the HE unit on

Ni-Co-P/NF electrodes, the FEs of about 99.0% and 98.4% have been achieved for HER and simultaneous HzOR, respectively (Fig. S37). Even after ADTs for 10000 cycles, the performance at the current density of 100 mA cm$^{-2}$ attenuates by only 11.6% according to the inset in Fig. 6d. The regular voltage fluctuations in the stability curve in Fig. 6d mainly come from the supplementations of $N_2H_4$ and the release of bubbles on the electrode (Fig. S38). Because of the large resistance (17.8 Ω) of AEM, the performance of Ni-Co-P/NF in HE unit has been operated with 50% iR compensation (Fig. S39), which shows a high current density of 500 mA cm$^{-2}$ at 0.498 V, higher than the reported capacities of HzOR-assisted HER in membrane-free or double-chamber electrolyzer (Table S5).

Based on the HzOR performance of Ni-Co-P/NF, a direct hydrazine fuel cell (DHzFC) equipped with Ni-Co-P/NF as the anode immersed in 1.0 M KOH + 0.1 M $N_2H_4$ electrolyte and 20 wt.% Pt/C as cathode immersed in 1.0 M KOH electrolyte was constructed. As shown in Fig. 6e, the DHzFC exhibits a peak power density of 96 mW cm$^{-2}$ at 0.75 V, which can drive the LED by two identical DHzFC in series (Fig. S40). Moreover, the DHzFC can run stably at the current densities of 1, 5, 10 and 15 mA cm$^{-2}$ for 4 h (Fig. S41), indicating its good operation stability at high output voltages. Furthermore, a self-powered $H_2$ production system has been constructed by integrating a DHzFC and the HE unit using Ni-Co-P/NF as the bifunctional catalysts (Fig. 6f). The self-made DHzFC successfully drives the HE unit for $H_2$ production, resulting in the production of a large number of bubbles (see the Supplementary Movie), and the $H_2$ evolution current is as high as 26.2 mA (Fig. 6g), corresponding to a hydrogen generation rate up to 19.6 mol h$^{-1}$ m$^{-2}$, proving the great potential in the utilization of self-powered hydrogen production during the non-carbon energy system.

In conclusion, an interface engineering strategy has been developed to synthesize a heterostructure nanoarray electrocatalyst (Ni-Co-P/NF) featuring CoP nanoparticles being uniformly distributed on the NiCoP nanowires, which acts as a bifunctional catalyst toward both HER and HzOR. This catalyst shows good HER/HzOR performances featuring especially low overpotential of 37 mV at 10 mAcm$^{-2}$ for HER, and −54 mV and 187 mV to reach 10 mA cm$^{-2}$ and 1000 mA cm$^{-2}$ for HzOR, better than most reported non-noble metal or noble metal catalysts. More importantly, the catalytic mechanism of HzOR has been probed in-depth: the instant recovery of the active MP component by $N_2H_4$ molecules during HzOR has been confirmed, and the HzOR has been thermodynamically identified to follow a new N-N bond breakage path at 0.2 V and above beyond the traditional PCET path, endowing Ni-Co-P/NF with the superior HzOR performance. Thus, the electrolyzer equipped with Ni-Co-P/NF as both anode and cathode catalysts offers as low as 0.498 V of applied voltage to reach 500 mA cm$^{-2}$ for $H_2$ production, greatly reducing the energy consumption for $H_2$ production. Equally importantly, the electrochemical utilization rate of $N_2H_4$ has been determined to be as high as 93% by tuning the $N_2H_4$ concentration. Also interestingly, the HzOR/HER electrolyzer can be powered by DHzFC equipped with Ni-Co-P/NF as anode catalyst to realize the self-powered $H_2$ production at the rate of 19.6 mol h$^{-1}$ m$^{-2}$. This work provides a remarkable performance HzOR catalyst and an unusual catalytic mechanism insight, benefiting the near future industrial hydrogen production at largely lowered energy consumption.

## Methods

### Materials

Hydrochloric acid (HCl), Urea (99 wt.%), Hydrazine monohydrate ($N_2H_4 \cdot H_2O$, 80 wt.%), Cobalt(II) nitrate hexahydrate (Co(NO$_3$)$_2$•6H$_2$O, 98.5 wt.%) and Nickel(II) nitrate hexahydrate (Ni(NO$_3$)$_2$•6H$_2$O, 98.0 wt.%) were purchased from Sinopharm Group Chemical Reagent Co., Ltd. Sodium hypophosphite (NaH$_2$PO$_2$, 99.0 wt.%) and Ammonium fluoride (NH$_4$F, 98 wt.%) were purchased from Aladdin. Sodium hydroxide (KOH, 90 wt.%) was purchased from Shanghai Titan

Scientific Co., Ltd. Acetone, Ethanol, Isopropanol were purchased from Shanghai Lingfeng. Nafion D-520 dispersion (5 wt.%) was purchased from Dupont China Holding Co., Ltd. Commercial 20 wt.% Pt/C and the carbon black (XC-72) were purchased from Shanghai HEPHAS Energy Equipment Co., Ltd. Nickel foam (thickness: 1.0 mm; aperture: 0.1 mm; porosity: 97.2%) was purchased from Cyber Electric Co., Ltd. Anion exchange membrane (AEM, FAB-PK-130) was purchased from Fumasep. Composite matrix was purchased from Changsha Sipulin. All materials were used as received without further purification.

### Preparation of Ni-Co-P/NF electrocatalysts

The three dimensional bimetallic phosphide nanowires heterostructure nanoarrays grown in nickel foam (Ni-Co-P/NF) were synthesized by facile two-step ways[23]. In a typical synthesis, the precursor nanowire arrays (Ni-Co-Pre/NF) were in situ grown on substrate of nickel foam through a hydrothermal method. Nickel foam was successively washed by acetone, 1 M HCl and deionized water with ultrasonic treatment for 10 min to get rid of the possible surface greasy dirt and oxide layer. Then, 3 mmol of $Co(NO_3)_2 \cdot 6H_2O$, 10 mmol of urea and 4 mmol of $NH_4F$ were dissolved into hydrothermal kettle lining with 60 mL of deionized water, and the pretreated nickel foam (2 cm × 3.5 cm) was put into and completely immersed in the solution. Afterwards, the hydrothermal kettle was maintained at 120 °C for 6 h. After cooled down naturally at room temperature, the substrate was then taken out and cleaned by ultrasonic treatment with DI water and ethanol several times before being fully dried. Similarly, the different proportions of Co precursors were prepared by the same steps with the addition of $Co(NO_3)_2 \cdot 6H_2O$ differing from 1 to 5 mmol.

Bimetallic phosphide is realized by a partial phosphide reaction of the above precursor in a tubular furnace. Typically, the substrate precursor prepared above and $NaH_2PO_2$ were placed at two separate locations of a long crucible with $NaH_2PO_2$ at the upstream side of the furnace and the precursor at the other side. Then, they were heated at 325 °C for 4 h with a heating speed of 2 °C $min^{-1}$ in $N_2$ atmosphere. The sample Ni-Co-P/NF was then obtained after cooling to room temperature in $N_2$ atmosphere. The mass loadings of Ni-Co-P/NF and Ni-Co-Pre/NF are about 4.42 and 4.80 mg $cm^{-2}$, respectively.

### Preparation of NiP/NF, CoP and Pt-C/NF electrocatalysts

The NiP/NF is prepared by the same method as Ni-Co-P/NF, only without adding $Co(NO_3)_2 \cdot 6H_2O$ to the precursor synthesis. The CoP is prepared by the same method as Ni-Co-P/NF, only without adding nickel foam to the precursor synthesis, which is dispersed on nickel foam during electrochemical tests. The mass loading of NiP/NF is about 2.27 mg $cm^{-2}$.

10 mg commercial 20 wt.% Pt/C was dispersed into 970 μL isopropanol, then 30 uL 10% Nafion solution was added, and after ultrasonic the ink solution is obtained. Then 100 μL ink is uniformly applied to 1 $cm^2$ nickel foam and obtain the Pt-C/NF. The mass loading of Pt-C/NF is about 2.00 mg $cm^{-2}$.

### Materials characterization

Scanning electron microscope (SEM) imaging was obtained using a FEI Magellan-400 field emission scanning electron microscope (5 kV). Transmission electron microscopy (TEM) patterns, high-resolution transmission electron microscopy (HRTEM), energy dispersive X-ray spectrometer (EDS) and corresponding EDS-mapping were recorded on a JEM-2100F field emission transmission electron microscope (200 kV). Spherical aberration-corrected HAADF-STEM and corresponding EDS-mapping measurements were taken on a JEM-ARM300F instrument. Powder X-ray diffraction (XRD) signals were performed at 4° $min^{-1}$ on a Rigaku D/Max-2550 V X-ray diffractometer with a Cu Kα radiation target (40 kV, 40 mA). X-ray photoelectron spectroscopy (XPS) data was tested on Thermo Fisher Scientific ECSAlab250 XPS spectrometer with monochromatic Al $K_\alpha$ radiation. Before

characterization, the catalysts have been etched by Ar ions. Raman and in situ Raman spectra were recorded on Raman S3 spectrometer (JY, Labram HR 800) with an excitation wavelength of 532 nm and inVia Qontorin Raman spectrometer (Renishaw). For the convenience of XRD, XPS, TEM and XAFS tests, fine powder samples were obtained by fully ultrasonic treated from NF substrates.

### Electrochemical measurements

All electrochemical performances were measured on a CHI 760E electrochemical workstation (CH instruments, Inc., Shanghai). Ag/AgCl electrode and carbon rod were used as reference and counter electrode, respectively. The Ag/AgCl electrode was stored in the saturated KCl solution and rinsed with deionized water before use. All the potential data reported in this work were converted to the reversible hydrogen electrode (RHE) according to the Nernst equation. All measurements related to HzOR were performed at 500 rpm to eliminate the effect of bubbles. The linear sweep voltammetry (LSV) measurements of HER were made at the scan rate of 5 mV $s^{-1}$, with 30% $iR_s$ compensation to eliminate the influence of solution impedance on the test results ($E_{corrected} = E_{measured} - 30\%$ I*$R_s$, where $E_{measured}$, I and $R_s$ are the experimentally measured potential, current and electrolyte internal resistance, respectively). The LSV measurements of HzOR were made at the scan rate of 50 mV $s^{-1}$, with 50% $iR_s$ compensation ($E_{corrrected} = E_{measured} - 50\%$ I*$R_s$). The Tafel plots were obtained according to the Tafel equation η = a log|j| + b. The accelerated durability tests (ADTs) were conducted for up to 10,000 cyclic voltammetry (CV) cycles, with the voltage range from −0.1 to 0.4 V (vs. RHE, in 1 M KOH and 0.1 M $N_2H_4$), and −0.3 to 0 V (vs. RHE, 1 M KOH), respectively. Electrochemical impedance spectroscopy (EIS) measurements were carried out in the frequency range from $10^{-1}$ to $10^{-6}$ Hz in different electrolytes. The electrochemical surface area (ECSA) data were calculated from CVs curves in a non-Faraday region (30–130 mV) at varied scan rates from 10 to 80 mV $s^{-1}$ and the double-layer capacitance ($C_{dl}$) values with potential range of CV tests from 30 to 130 mV. The in situ Raman test was carried out in 1 M KOH or 1 M KOH + 0.1 M $N_2H_4$, using the catalysts cut into 1 cm × 1 cm as work electrode and Ag/AgCl electrode as reference electrode. The electrode potential was set from −0.05 V to 0.5 V and then was set at 0.1 V for 60 min, during which Raman test was carried out. The faradaic efficiencies of HER were calculated by the amount of charge transfer and the volume of $H_2$ collected by the gas gathering unit. Theoretical $H_2$ volume was obtained based on Faradaic law of electrolysis. Unless otherwise specialized, all the potentials are referred to the RHE and the data is not processed by iR compensation.

### Products analysis

The gas products from the cells were collected and examined by gas chromatography (GC, Shimadzu GC-2014C) equipped with a thermal conductivity detector (TCD) operated at a fixed temperature of 25 °C according to the internal standard method. Meanwhile, the theoretical volume of evolved gas can be calculated by the equation:

$$V = I \times t \times V_m / (n \times F)$$

where the V is the theoretical volume (mL) of evolved gas after electrolysis for a certain time (t) at a fixed current (I), $V_m$ is the gas molar volume (22.4 mol $L^{-1}$), n is the number of electrons transferred ($n = 2$ for HER, $n = 4$ for HzOR), F is the Faraday constant (96485 C $mol^{-1}$). The Faradaic efficiency can be estimated according to the ratio of the measured to the theoretical gas volume.

### Calculation of the utilization rate

The gas products from cells of the HE unit were collected by drainage collection method. In the anode, the volume of $N_2$ ($V_C/2$) produced by the electrochemical oxidation of $N_2H_4$ can be inferred from the $H_2$

volume ($V_C$) of cathodic HER, and the remaining gas volume ($V_A$-$V_C$/2) is corresponds to the total volume of $H_2$ and $N_2$ generated by $N_2H_4$ spontaneously decomposed. Thus, the $N_2$ volume from $N_2H_4$ spontaneously decomposed in the anode is ($V_A$-$V_C$/2)/3. Therefore, the utilization ratio (η, proportion of hydrazine used for oxidation) is calculated by the equation:

$$\eta = 3V_C/2(V_A + V_C)$$

where the $V_A$, $V_C$ are the gas volume produced in anode and cathode, respectively.

## DFT calculation details

DFT calculations were performed using the Vienna Ab Initio Simulation Package (VASP)[51,52]. The Perdew-Burke-Ernzehof (PBE) functional and the projector augmented wave (PAW) method were applied in our calculation[53]. The NiCoP-CoP heterostructure was constructed by integrating the NiCoP(111) and CoP(011) surfaces into contact with each other, and the lattice mismatch is around rather low at 4%. Vacuum thickness over 12 Å was applied. A kinetic energy cut-off of 520 eV and gamma K-point only were used during the structural optimization. The maximum force exerted on each atom was relaxed to less than 0.02 eV Å$^{-1}$ and the convergence criteria of energy difference is 10$^{-5}$ eV. Dispersion correction by Grimme's DFT-D3 scheme was adopted to describe the van der Waals (vdW) interactions[54]. HzOR proceed through two possible mechanisms: the traditional path(I) and N-N breakage path(II).

**Path I:** $N_2H_4 \rightarrow {}^*N_2H_4 \rightarrow {}^*N_2H_3 \rightarrow {}^*N_2H_2 \rightarrow {}^*N_2H \rightarrow {}^*N_2 \rightarrow N_2$
$N_2H_4 + {}^* \rightarrow {}^*N_2H_4$
${}^*N_2H_4 \rightarrow {}^*N_2H_3 + H^+ + e^-$
${}^*N_2H_3 \rightarrow {}^*N_2H_2 + H^+ + e^-$
${}^*N_2H_2 \rightarrow {}^*N_2H + H^+ + e^-$
${}^*N_2H \rightarrow {}^*N_2 + H^+ + e^-$
${}^*N_2 \rightarrow N_2 + {}^*$

**Path II:** $N_2H_4 \rightarrow {}^*N_2H_4 \rightarrow 2{}^*NH_2 \rightarrow 2{}^*NH \rightarrow 2{}^*N \rightarrow {}^*N_2 \rightarrow N_2$
$N_2H_4 + {}^* \rightarrow {}^*N_2H_4$
${}^*N_2H_4 + {}^* \rightarrow 2{}^*NH_2$
$2{}^*NH_2 \rightarrow 2{}^*NH + 2H^+ + 2e^-$
$2{}^*NH \rightarrow 2{}^*N + 2H^+ + 2e^-$
$2{}^*N \rightarrow {}^*N_2 + {}^*$
${}^*N_2 \rightarrow N_2 + {}^*$

The asterisk (*) represents the reaction surfaces in the models NiCoP (111), CoP (011), and NiCoP (111) | CoP (011). "*$N_2H_4$" "*$N_2H_3$" "*$N_2H_3$" "*$N_2H_3$" "*$N_2H_3$" "*$NH_2$" "*$NH$" "*$N$" denote the models with the corresponding adsorbed intermediates residing on the reaction surfaces. The Gibbs free energy diagrams of intermediates were calculated based on computational normal hydrogen electrode (NHE) model proposed by Norskov et al.[55]. For each elementary step, ΔG was evaluated by the following equation:

$$\Delta G = \Delta E + \Delta ZPE - T\Delta S + \Delta G_U + \Delta Gp_H + \Delta G_{field}$$

where ΔE, ΔZPE and ΔS are the reaction energy, the zero-point energy contribution, and entropy change, respectively. T is temperature (set to 298.15 K). The ΔS values of gas phases $H_2$, $N_2$ and $N_2H_4$ were obtained from the NIST-JANAF thermodynamics table, and ΔZPE was calculated by the vibration contribution of all adsorbed species[56]. The free energy contributed by the electrode potential (relative to NHE) was assessed with the equation $\Delta G_U = -neU$, where n is the number of electrons transferred. $\Delta G_{pH}$, related to OH$^-$ in electrolyte, is the correction to the H$^+$ free energy, which can be calculated through $\Delta G_{pH} = 2.3k_BT{*}pH$, and the $k_B$ is the Boltzmann constant. $\Delta G_{field}$ is the free-energy correction resulted from the electrochemical double layer related to the anion and cation in electrolyte.

Ab initio molecular dynamics (AIMD) simulations were performed with explicit treatment of the water environment at room temperature, which contains 15 $H_2O$ molecules and one OH. All atoms are statically relaxed and stable at room temperature. The NVT ensemble using a Nose–Hoover thermostat was performed.

## Data availability

All data are available in the manuscript, the supplementary materials and from the authors on request. Source data of the manuscript are provided as a Source Data file. Source data are provided with this paper.

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

## Acknowledgements

We sincerely appreciate the support for this research by National Natural Science Foundation of China (52172110), Key Research Program of Frontier Sciences, Chinese Academy of Sciences (ZDBS-LY-SLH029), the "Scientific and Technical Innovation Action Plan" Hong Kong, Macao and Taiwan Science & Technology Cooperation Project of Shanghai Science and Technology Committee (21520760500), and the BL14W1 beamline of Shanghai Synchrotron Radiation Facility (SSRF).

## Author contributions

L.Z., J.S. and X.C. designated the idea of this work. L.Z., G.M. and C.C. performed the experiments. J.H. performed theoretical calculations. T.W., H.T., Y.C., F.K., and Z.C. helped with the material characterization. L.Z. and X.C. wrote the whole manuscript. J.S. and X.C. supervised the project, revised the manuscript, and commented on it.

## Competing interests

The authors declare no competing interests.
