## [Peer review file · Nature Communications]

REVIEWER COMMENTS

Reviewer #2 (Remarks to the Author):

In this work, the authors reported a bimetallic phosphide for HER/H₂OR electrocatalysis, and detailedly studied the corresponding electrocatalytic mechanism. Impressively, the H₂OR catalytic mechanism has been revealed in depth---the extremely low reaction energy barrier and the recovery of active site metal phosphide by N₂H₄ molecules, leading to the significant enhancement of the electrocatalytic performance. The result seems interesting. This reviewer think that this manuscript could be publication in the NC after addressing the following issues.

1. Why are the nanowires separated from nanoparticles in the TEM image in Fig. 1c? It is contradictory with the statements that nanoparticles are uniformly dispersed on the surface of nanowires.
2. The loading amount of electrocatalyst is an important factor for evaluation of its performance. The loading amount of Ni-Co-P/NF should be provided when comparing with Pt-C/NF.
3. XPS results analysis of the Ni-Co-P/NF-inactivated proved the valence states of the catalyst surface, it is suggested to characterize the catalyst bulk phase components by other characterization methods to verify the evolution of catalyst structures during electrolysis.
4. The recovery of the active site MP of the transition metal phosphide catalyst (TMPs) during H₂OR ensures the durability for a long time. Is this mechanism universal for similar reactions ?
5. The author claimed the electroless “spontaneous decomposition” in the anode chamber for the first time, which is however obviously unfavorable to construction of the HE unit. I suggest the authors to carefully clarify this point.
6. The authors estimated the ratio of MPO_x/MP in the catalyst at Line328, Page14 according to the XPS results. How do you get this result since XPS only represents the materials surface information.
7. The unit of the X axis “V vs. RHE” of Fig. 4h should be changed. Please check the equation 2, Line321, Page14, the total charge numbers on both sides of the equation cannot be conserved.
8. Many mistakes in reference section need to be carefully revised before publication.

Reviewer #3 (Remarks to the Author):

The manuscript describes Co-NiP catalysts for hydrazine electrooxidation in detail and also show the called up process in 2 compartment electorlyzer. Additionally, the fuel cell performance is also tested.

Although the work is of great interest in the current era as the world is poised towards green hydrogen production. I am somehow disappointed with the paper as a whole because it lacks several important details. I am therefore recommending a rejection. My specific comments are as follows:

1. Authors fail to provide the novelty aspect in their catalytic system because the CO-NiP is a well known catalyst.
2. If hydrogen generation is the aim, why decomposition should be a limitation for electrolysis. Besides, it has been mentioned that the low concentration of hydrazine prevents decomposition, how did authors confirm it?
3. Reference 12 is not related to catalytic decomposition.
4. Ni-Co-P heterostructure were prepared over Nickel foam so from where separated Co-P nanoparticles are forming?
5. Under TEM explanation on page 6, What is the relevance ultrasound is?
6. What is the meaning of Tafel mechanism? They should also compare the exchange current density and comment on the kinetics.
7. On page 10, why the stability is being measured at such a low voltage which is producing less than 10 mA/cm² of current. If the claim is to produce high currents like 500-1000 mA/cm², stability at 100 mA/cm² should be checked.
8. Figure S14 is not explained properly as the authors are also estimating the yield which is not even explained. What do they mean by theoretical yield?
9. In Figure 5d, why the experiment was not performed with only KOH. Otherwise the idea of oxidation and deactivation will not be proven.
10. In Figure 5e, the CV has the reversibility because the voltage is applied and reversed. How come just stopping the voltage application is reverting the catalyst? What is the mechanism?
11. Authors have not done a proper literature survey. Some important papers such as ACS Catalysis 11 (22), 14000-14007, 2021.

Reviewer #2 (Remarks to the Author):

Comments:

In this work, the authors reported a bimetallic phosphide for HER/H₂O₂ electrocatalysis, and detailedly studied the corresponding electrocatalytic mechanism. Impressively, the H₂O₂ catalytic mechanism has been revealed in depth---the extremely low reaction energy barrier and the recovery of active site metal phosphide by N₂H₄ molecules, leading to the significant enhancement of the electrocatalytic performance. The result seems interesting. This reviewer think that this manuscript could be publication in the NC after addressing the following issues.

Response: Thank you very much for your positive recommendation. We appreciate the reviewer's time and efforts in reviewing the manuscript. We have supplemented a number of experiments and revised the manuscript according to the editors' and reviewers' suggestions, and we hope the revision have addressed your valuable comments. Please find our point-by-point responses below.

1. Why are the nanowires separated from nanoparticles in the TEM image in Fig. 1c? It is contradictory with the statements that nanoparticles are uniformly dispersed on the surface of nanowires.

Response: Thank you very much for the question. The catalyst (Ni-Co-P/NF) was loaded on the nickel foam (NF), and the magnetic character of NF would interfere with the electric beam of TEM, so the catalyst is always removed from the NF by ultrasound prior to TEM observation, which has also been reported in literature (*Adv. Funct. Mater.*, 2016, 26, 4661). In this work, the catalysts were fully ultrasonically removed from the NF before TEM characterization, resulting in the separated nanowires and discrete nanoparticles. The relevant information is shown in the revised Manuscript (**line 582, Page 24**).

2. The loading amount of electrocatalyst is an important factor for evaluation of its performance. The loading amount of Ni-Co-P/NF should be provided when comparing

with Pt-C/NF.

Response: Thanks a lot for the kind suggestion. The loading amount of the electrocatalyst has been quantified. The mass loading amounts of Ni-Co-P/NF, Ni-Co-Pre/NF, NiP/NF and Pt-C/NF are 4.42, 4.80, 2.27 and 2.00 mg cm⁻², respectively, which have been supplemented in the revised Manuscript (**Page 23**).

3. XPS results analysis of the Ni-Co-P/NF-inactivated proved the valence states of the catalyst surface, it is suggested to characterize the catalyst bulk phase components by other characterization methods to verify the evolution of catalyst structures during electrolysis.

Response: Thank you very much for the constructive suggestion. First, the XPS results give the composition changes of approximately 10 nm in depth on the catalyst surface. However, before characterization, the catalysts were etched by Ar ions to remove a layer of oxides and impurities from the surface, and the corresponding statement has been given on **line 579, Page 24** in the revised Manuscript. Therefore, the XPS results can provide the composition changes at a certain depth (bulk phase) (*Nat. Commun.*, 2022, 13, 2916).

In addition, compared with XPS, the EDS can better show the changes of the whole catalyst as reported in literature (*Energy Environ. Sci.*, 2022, Accepted Manuscript). The micro-area on the surface of samples Ni-Co-P/NF and Ni-Co-P/NF-inactivated have been measured by the EDS of TEM. The supplemented data have been presented in **Fig. S23** (see below), in the revised Supporting Information, showing the consistent conclusion with the XPS result. After deactivation, the percentage of oxygen in the catalyst Ni-Co-P/NF-inactivated will increase significantly, verifying the increase in the relative amount of metal phosphorus oxide. Relevant discussion has been supplemented in **line 343-349, Page 15** in the revised Manuscript.

Fig. S23. TEM-EDS images of Ni-Co-P/NF (a) and Ni-Co-P/NF-inactivated (b); (c) the atomic percentages of Ni-Co-P/NF and Ni-Co-P/NF-inactivated.

Moreover, the *in situ* Raman data can also qualitatively demonstrate the increase of oxygen amount (P-O) in the bulk phase, thus manifesting the change of the bulk phase components (*Nat. Commun.*, 2022, 13, 2916; *Chem. Soc. Rev.*, 2021, 50, 3519-3564). The Raman spectra before and after the deactivation of catalysts have been supplemented as shown in **Fig. S24** (see below) in the revised Supporting Information, and a new peak (corresponding to the stretching bond of P-O) could be found at 1020 cm^{-1} when the voltage was about 0.2 V, and its intensity increased at the elevated voltage, while that of M-P decreased correspondingly, verifying the transformation from MP to MPO_x under voltage in equation (1). The relevant discussion is shown on **line 352-357**, **Page 15** in the revised Manuscript.

Fig. S24. *In situ* electrochemical Raman spectra of Ni-Co-P/NF in 1 M KOH at varied applied potentials.

4. The recovery of the active site MP of the transition metal phosphide catalyst (TMPs) during HzOR ensures the durability for a long time. Is this mechanism universal for similar reactions?

Response: Thanks a lot. In fact, there is a premise for this mechanism: the starting potential of HzOR should be lower than the electrochemical oxidation potential of the catalyst itself (**Fig. 5a**). It is necessary that the reductibility of the substrate should be stronger than that of the catalyst for oxidizing reactions. Such oxidation reactions include the sodium borohydride oxidation reaction (BOR). The BOR performance of MP was initially explored using the scheme similar to that in this work, and the preliminary work has been carried out as shown in **Fig. S30** (see below) in the revised Supporting Information. The initial results show that this mechanism is applicable to BOR at least. Relevant discussion is shown in **Page 18 (line 414-420)** in the revised manuscript.

Fig. S30. The mutual promotion relationship between metal phosphide and borohydride. (a) CV curves with or without NaBH₄ addition in electrolyte; (b) LSV curves of NiCoP toward BOR initially and after over-oxidation; (c) The stability measurement at about 100 mA cm⁻² of Ni-Co-P/NF for BOR in 1 M KOH + 0.1 M NaBH₄.

5. The author claimed the electroless “spontaneous decomposition” in the anode chamber for the first time, which is however obviously unfavorable to construction of the HE unit. I suggest the authors to carefully clarify this point.

Response: Thank you very much for the kind reminding. The discovery of this electroless “spontaneous decomposition” phenomenon of hydrazine makes it necessary to reconsider the utilization rate (η) of N₂H₄ when designing the hydrazine oxidation assisted hydrogen evolution unit in the future. In fact, both the electroless “spontaneous decomposition” and electrochemical oxidation of hydrazine are co-existing during the anode hydrazine oxidation, and the former will reduce the utilization rate of N₂H₄ which is undesirable in HzOR assisted HER unit. While the electroless “spontaneous decomposition” phenomenon of hydrazine is usually ignored. In this work, this problem has been preliminary discussed. As mentioned in **Page 19**, reducing the concentration of N₂H₄ can weaken the “spontaneous decomposition”, but the reduction of N₂H₄

concentration will have an adverse effect on the current density of HER, so there should be an optimum between these two factors, which reminds the subsequent researchers to fully consider the impact of the concentration of N_2H_4 instead of blindly increasing its concentration. Secondly, it gives guidance on how to design the catalyst. For example, in the design of hydrazine oxidation catalyst in the future, it can be considered to weaken its HER performance. Because the root of the “spontaneous decomposition” is the excellent activity of the catalyst for HER and HzOR, resulting in their extremely low over potential, so that HER and HzOR can occur spontaneously at anode. Due to this phenomenon, the strategy of designing HER/HzOR bifunctional catalyst should be carefully considered in the future.

6. The authors estimated the ratio of MPO_x/MP in the catalyst at Line328, Page14 according to the XPS results. How do you get this result since XPS only represents the materials surface information.

Response: Thanks a lot. Indeed in this work, the ratio of MPO_x/MP was estimated to characterize the surface chemical states, not the interior, of the catalysts by XPS. We are sorry for the inaccurate statement, and now it has been corrected as “...it can be found that the ratio of MPO_x to MP significantly increased on the surface of catalysts,” in **line 341-342, Page15** in the revised Manuscript. In addition, it is common to characterize the change of valence state of catalysts during the electro catalytic reaction by XPS, because the active sites are distributed on the surface of the catalysts (*Angew. Chem. Int. Ed.*, 2021, 60, 7297; *Nat. Commun.*, 2022, 13, 2916). So the ratio of peak area from the XPS spectra (**Fig. S22c, d**) was calculated to directly reflect the relative proportion of MP and MPO_x in the surface of catalysts.

7. The unit of the X axis “V vs. RHE” of Fig. 4h should be changed. Please check the equation 2, Line321, Page14, the total charge numbers on both sides of the equation cannot be conserved.

Response: Thanks very much for your kind reminding, and we are very sorry for this mistake. The labeling of **Fig. 4h** and **equation (2) in Page15** have been corrected in the

revised Manuscript.

8. Many mistakes in reference section need to be carefully revised before publication.

Response: Many thanks. We have carefully checked the reference section, and revised the corresponding **References (3, 6, 11, 22, 39, 50, 51)** in the revised Manuscript.

Reviewer #3 (Remarks to the Author):

Comments:

The manuscript describes Co-NiP catalysts for hydrazine electrooxidation in detail and also show the called up process in 2 compartment electrolyzer. Additionally, the fuel cell performance is also tested. Although the work is of great interest in the current era as the world is poised towards green hydrogen production. I am somehow disappointed with the paper as a whole because it lack several important details. I am therefore recommending a rejection. My specific comments are as follows:

Response: Thank you very much for your comments. We appreciate the reviewer's time and efforts in reviewing the manuscript. We have supplemented numbers of experiments and revised the manuscript according to your questions and suggestions, and we hope the major revision have addressed your valuable comments. Please find our point-by-point responses below.

1. Authors fail to provide the novelty aspect in there catalytic system because the Co-NiP is a well known catalyst.

Response: Thanks a lot. Indeed the catalyst Co-NiP is a well-known catalyst and has been widely reported in HER and other reactions, while there are few reports on catalyzing HzOR by using Co-NiP, and moreover, the catalytic mechanism of phosphide system for HzOR has not been probed in-depth. In this work, we constructed the heterostructure <CoP|NiCoP> with the CoP nanoparticles uniformly distributed on the NiCoP nanowires (**Fig. 1b and 1f**), and obtained the excellent high current density (above 1600 mAcm^{-2}) toward HzOR (**Fig. 4a and Fig. 4d**) by overcoming surface mass transfer through interface engineering (**Fig. 3c and Fig. 4c**). We further demonstrate that the excellent durability of Ni-Co-P/NF during HzOR (**Fig. 4g**) comes from the durable active site (metal phosphide) due to the presence of N_2H_4 , and the electronic structure changes across the heterogeneous interface (**Fig. 5**). The novelty of the catalytic system of Ni-Co-P/NF toward HzOR is described as shown in **line 70-74**,

Page 4.

The stability of HzOR under high current density is one of the biggest challenges (*Angew. Chem. Int. Ed.*, 2021, 60, 5984–5993; *Nat. Commun.*, 2020, 11, 1853; *Energy Environ. Sci.*, 2022, 15, 3246-3256; *Angew. Chem. Int. Ed.*, 2022, 61, e202203929), and we have checked the stability performances of various reported catalysts in **Tale S4**, indicating that the stability for most reported catalysts can only be maintained for tens of hours at a low current density, which is undesirable for the hydrogen production process in future industrial applications. The catalyst designed in the present work (Co-NiP) exhibits excellent stability during HzOR at a high current density of 100 mAcm⁻² (**Fig. 4g**) and shows no significant decline in the end of test for 100 h, which is significantly better than the literature reports as far as we know.

Most importantly, the evolution of metal phosphide catalyst in the process of HzOR has been discussed in detail. In general, similar to OER, the transition metal-based catalysts are easy to be oxidized under the positive voltage (*Nat. Commun.*, 2022, 13, 2916), and gradually lose the HzOR activity. Therefore, the stability of most of catalysts is rather poor. It is noticeable that the catalyst (Ni-Co-P/NF) prepared in this work not only has high current density (**Fig.4a**), but also possesses excellent stability (**Fig. 4g**). Through *in situ* Raman spectra and quasi in-situ XPS analysis, the evolution of catalyst in the reaction process can be monitored (*Angew. Chem. Int. Ed.*, 2021, 60, 7297–7307). Phosphide will also be oxidized under the applied voltage (**Fig. 5b**). If the oxidation conversion continues, the performance of the catalyst (Ni-Co-P/NF) will become deteriorated (**Fig. S21**). However, hydrazine itself can be used as a reducing agent to reduce the over oxidation of metal phosphide and avoid the influence of anode voltage on the catalyst, resultantly endowing the active metal phosphide with excellent stability. All the above novelties are believed to be of significance to the community.

2. If hydrogen generation is the aim, why decomposition should be a limitation for electrolysis. Besides, it has been mentioned that the low concentration of hydrazine prevents decomposition, how did author so confirm it?

Response: Many thanks. The decomposition of N₂H₄ (N₂H₄→N₂+2H₂) will produce

N_2 and H_2 simultaneously, which requires complicated and high-cost processes to purify H_2 (*J. Phys. Chem., C*, 2011, 115, 47, 23261–23266). Fortunately, H_2 production through HER assisted by HzOR (**path 2 in Fig. 6b**) can be performed by applying exchange membrane to separate anode nitrogen from cathode H_2 and obtain the high-purity H_2 (*Nat. Commun.*, 2021, 12, 4182). However, as shown in **Fig. S31**, the direct “decomposition” of N_2H_4 (**path 1 in Fig. 6b**) is inevitable during HzOR in the anode chamber. Therefore, with consideration of the purity of the H_2 , the **path 2 in Fig. 6b** is defined here as the effective oxidation of hydrazine while the **path 1 in Fig. 6b** is the undesirable “decomposition” which is a limitation for electrolysis. Thus improving the utilization of N_2H_4 for HzOR electrolysis is essential for the performance of electrocatalysts.

As for the influence of N_2H_4 concentration on the “spontaneous decomposition” process, the proportions of N_2H_4 used for “decomposition” and electrochemical oxidation in the anode can be calculated by detecting the gas products of different concentrations of N_2H_4 in the oxidation process. If 1 mole of hydrazine is directly decomposed (in **path 1 in Fig. 6b**), 2 moles of H_2 and 1 mole of N_2 will be produced in the anode. From the H_2 volume (V_C) of cathodic HER in **path 2 in Fig. 6b**, we can calculate the volume of N_2 ($V_C/2$) generated by the electrochemical oxidation of N_2H_4 in the anode, and the remaining gas ($V_A - V_C/2$) at the anode corresponds to the total volume of H_2 and N_2 generated by spontaneously decomposed N_2H_4 since both electrochemical oxidation and “spontaneous decomposition” of N_2H_4 take place at anode. Thus the proportion of N_2H_4 used for oxidation and “decomposition” at different N_2H_4 concentrations can be calculated. (Relevant discussion has been supplemented in **Page 25** in the revised Manuscript). It can be seen evidently from the **Fig. 6c** that with the increase of N_2H_4 concentration, the percentage of H_2 in the anode gas becomes higher, indicating that the “decomposition” ratio becomes higher. In addition, the effect of N_2H_4 concentration on its decomposition has also been reported in literature (*J. Am. Chem. Soc.*, 2009, 131, 50, 18032–18033) that the reaction rate decreases in the time course of reaction because of the decreased N_2H_4 concentration. Based on above considerations, we confirm that the low concentration of hydrazine prefers the

“decomposition” prevention.

3. Reference 12 is not related to catalytic decomposition.

Response: Thank you very much for your careful review. We have carefully checked the whole references, and replaced Reference 12 (in original edition) as the true reference (**now Reference 11**) in the revised Manuscript.

4. Ni-Co-P heterostructure were prepared over Nickel foam so from where separated Co-P nanoparticles are forming?

Response: Thank you very much for the question. The CoP nanoparticles come from the precursor $\text{Co}(\text{CO}_3)_{0.5}(\text{OH})$ as shown in **Fig. S3c**. In the phosphating process, and at elevated temperatures, the H_2O and CO_2 bubbles were produced at the surface of Ni-Co-P nanowires, then the PH_3 gas reduces the precursor of the nanowires to the CoP on the surface of the nanowire, which is consistent with the literature reports (*Adv. Funct. Mater.*, 2012, 22, 861–871; *Adv. Mater.*, 2017, 29, 1602441). Under the effect of gas etching and temperature rise, CoP species on the surface of nanowires gradually agglomerates and forms nanoparticles. As shown in the **inset of Fig. 1b**, the CoP nanoparticles are uniformly distributed on the surface of the NiCoP nanowires forming a heterogeneous interface in the final prepared catalysts.

During the TEM characterization, Ni-Co-P powder samples were obtained by scraping them from the nickel foam (NF) to avoid the influence of magnetic NF substrate, followed by thorough ultrasonic treatments as described in **Page 24**, resulting in the separation of CoP nanoparticles from NiCoP nanowires during ultrasound as shown in **Fig 1c**.

5. Under TEM explanation on page 6, What is the relevance ultrasound is?

Response: Thanks a lot for your question. Due to the problem by the magnetism of NF substrate for TEM characterization, we first prepared the catalyst powder samples by scraping them from the surface of NF substrates by ultrasound to characterize the microstructure, and this is the “relevance ultrasound” on **page 6**, which is also reported

in literatures (*Adv. Funct. Mater.*, 2016, 26, 4661; *Adv. Energy Mater.*, 2020, 10, 1903891). To describe it more accurately, the relevant discussion has been corrected in **line 122, Page 6** in the revised Manuscript.

6. What is the meaning of Tafel mechanism? They should also compare the exchange current density and comment on the kinetics.

Response: Thanks a lot for your question and the suggestion. The generally accepted reaction mechanism of HER in alkaline solution follows either Volmer-Heyrosky or Volmer-Tafel step, in the Tafel step two adsorbed hydrogen atoms combine on the surface of the electrode to give H₂ (*Chem. Soc. Rev.*, 2014, 43, 6555-6569).

In this work, the Tafel slope of Ni-Co-P/NF is 33.1 mV dec⁻¹, which is close to 29 mV dec⁻¹, indicating the fast discharge reaction and the Volmer-Tafel mechanism as the HER pathway, in which the recombination of chemisorbed hydrogen atoms (Tafel step) is the rate-limiting step. (*Chem. Soc. Rev.*, 2015, 44, 2060-2086; *Nat. Nanotechnol.*, 2017, 12, 441-446) To describe it more accurately, the relevant discussion has been corrected in **Page 10 (line 210-215)** in the revised Manuscript. At a relatively low Tafel slope, the HER rate of Ni-Co-P/NF will increase rapidly with increasing overpotential, which favors practical applications. By extrapolating the Tafel plots, the exchange current density of Ni-Co-P/NF was obtained (1.197 mA cm⁻²). This value is much higher than those of other catalysts, even higher than that of Pt/C (0.638 mA cm⁻²), indicating superior intrinsic electrocatalytic activity of the designed Ni-Co-P/NF.

According to your suggestion, the data related to exchange current density (J_0) have been further calculated from the intercept of the linear region of Tafel curves and supplemented in **the corrected Fig. 3b** in the revised Manuscript. The J_0 for Ni-Co-P/NF is 1.197 mA cm⁻², which is about twice of that of Pt-C/NF (0.638 mA cm⁻²), meaning that the system can output large currents even at very low overpotential and the HER can be easily activated with fast electrode kinetics, which is consistent with the literatures (*Nat. Nanotechnol.*, 2017, 12, 441-446; *Chem. Soc. Rev.*, 2015, 44, 2060-2086). Typically, the J_0 value is expected to be proportional to catalytically active surface area (*J. Am. Chem. Soc.*, 2014, 136, 13, 4897-4900), indicating that the high

catalytic activity of Ni-Co-P/NF comes from the large active area by interface engineering. Relevant discussion has been supplemented in **Page 10 (line 215-221)** in the revised Manuscript.

Fig. 3b. Tafel plots and exchange current densities.

7. On page 10, why the stability is being measured at such as low voltage which is producing lesser than 10 mA/cm² of current. If the claim is to produce high currents like 500-1000 mA/cm², stability at 100 mA/cm² should be checked.

Response: Thanks a lot. According to your suggestion, we have further provided the stability measurement results at the current density of 100 mA cm⁻² for about 100 h as shown in **Fig. S12** (see below) in the revised Supporting information. Relevant discussion has been supplemented in **Page 10 (line 234-235)** in the revised Manuscript.

Fig. S12. The stability measurement of Ni-Co-P/NF for HER in 1 M KOH at the constant potential of -160 mV with the current density of about 100 mA cm⁻².

8. Figure S14 is not explained properly as the authors are also estimating the yield which is not even explained. What do they mean by theoretical yield?

Response: Thank you very much for the question. Figure S14 (**now Fig. S15b**) shows the Faradaic efficiency (FE) of Ni-Co-P/NF toward HER, and the time on the horizontal axis is directly related to the amount of charge transferred between anode and cathode. From this amount of charge, it can be inferred that the theoretical hydrogen production amount is the gas volume generated when all transferred charges in the system are from HER. The theoretical gas yield is calculated by the equation:

$$V = I \times t \times V_m / (n \times F)$$

Where the V is the theoretical volume (mL) of evolved gas after electrolysis for a certain time (t) at a fixed current (I), V_m is the gas molar volume (22.4 mol L^{-1}), n is the number of electrons transferred (n = 2 for HER, n = 4 for HzOR), F is the Faraday constant (96485 C mol^{-1}). The measured N_2 or H_2 yield is calculated by gas chromatography (GC) equipped with a thermal conductivity detector (TCD) according to the internal standard method. (*Adv. Energy Mater.*, 2015, 5, 1401879; *Nat. Commun.*, 2021, 12, 4182) The Faradaic efficiency (FE) can be estimated according to the ratio of the measured to the theoretical gas yield (Relevant discussion has been supplemented in **Page 25** in the revised Manuscript).

9. In Figure 5d, why the experiment was not performed with only KOH. Otherwise the idea of oxidation and deactivation will not be proven.

Response: Thank you very much for the constructive comment. It is true that the experiment performed with only KOH is the key to prove the oxidation and deactivation of catalysts. To make the issues clearer, the *in situ* Raman spectra of Ni-Co-P/NF at varied applied voltages in 1 M KOH has been supplemented shown in **Fig. S24** (see below) in the revised Supporting Information. It has been found that a new peak, corresponding to the stretching bond of P-O (*Adv. Energy Mater.*, 2022, 12, 2201141; *Sci. Rep.*, 2018, 8, 12966), appeared at 1020 cm^{-1} when the voltage was about 0.2 V, and the Raman peak intensity increased with the voltage elevation from 0.2 V to 0.5 V, while the peak intensity of M-P (at around 400 cm^{-1}) decreases gradually with the

increase of voltage, verifying the transformation from MP to MPO_x under voltage in **equation (1)** ($MP + OH^- \leftrightarrow MPO_x + H_2O + e^-$). Relevant discussion has been supplemented in **Page 15 (line 352-357)** in the revised Manuscript.

Fig. S24. *In situ* electrochemical Raman spectra of Ni-Co-P/NF in 1 M KOH at varied applied potentials.

10. In Figure 5e, the CV has the reversibility because the voltage is applied and reversed. How come just stopping the voltage application is reverting the catalyst? What is the mechanism?

Response: Thanks a lot for your comment and question. From **Fig. 5a**, there are two types of oxidation peaks: A_1 and A_2 , and the oxidation peak of N_2H_4 (A_2) at lower potential indicates that N_2H_4 oxidization prior to the oxidization of the catalyst (Ni-Co-P/NF). Therefore, once the catalyst is oxidized during the reaction, N_2H_4 of strong reducibility will gradually reduce the catalyst to its pristine state, according to **equation (2)** ($MPO_x + N_2H_4 \rightarrow MP + N_2 + H_2O$), where the N_2H_4 used as the H source and electron donor (*Angew. Chem. Int. Ed.*, 2015, 54, 841–845), and such a reduction is directly associated with the removal of oxyanions intercalated in the lattice of metal phosphorus oxide (*Inorg. Chem.*, 2019, 58, 8, 4989–4996; *Nat. Commun.*, 2022, 13, 2916). During the HER process under the reduction potential, the oxides can be converted into phosphides (*Angew. Chem. Int. Ed.*, 2020, 59, 21106–21113), while in 1.0 M KOH + 0.1 M N_2H_4 electrolyte, the excellent reducibility of N_2H_4 makes the external reduction voltage no longer necessary, which can be proved by the disappearance of the reduction peak B_1 in **Fig. 5a**. That means, hydrazine itself can

reduce metal phosphorus oxides and maintain metal phosphide to be catalytically active, which is reasoned to be the mechanism of catalyst reverting and the source of excellent durability of metal phosphide catalyst during the HzOR.

11. Authers have not done a proper litearture survey. Some important papers such as ACS Catalysis 11 (22), 14000-14007, 2021.

Response: Thank you very much for supplying the paper. We have carefully checked the whole manuscript (ACS Catalysis 11 (22), 14000-14007, 2021) and cited the relevant papers in the **Reference (51)**.

Thanks again for your time and effort. Following your suggestions, we have supplemented major functional and pharmacological studies as presented above. We hope that the major revision could well-address your concern.

REVIEWER COMMENTS

Reviewer #2 (Remarks to the Author):

The authors have properly addressed all my concerns, this reviewer thus recommended its acceptance on nature communications.

Reviewer #4 (Remarks to the Author):

In the manuscript "Active site recovery and N-N bond breakage during hydrazine oxidation boosting the electrochemical hydrogen production", Zhu and co-workers have investigated the use of a new nanostructured interface as alternative catalysis to boost the HER/H₂OR electrocatalysis. The approach is clearly interesting and the topic is very timely as green-hydrogen production is high in nowadays challenges. The manuscript is well-written, however, is rather lengthy and the conclusions are diluted in the text. The computational part, for example, is using a rather simple model and the computational conclusions are rather detached from the rest of the text. I would recommend the authors integrate better the different aspects of the study before publication. Moreover, the following points should be addressed:

- 1) how did the authors obtain the interface used as a computational model? It is not clear how realistic and representative the model is.
- 2) how stable is the structure when in contact with the electrolyte? Is there leakage of Ni or Co?
- 3) how representative are the adsorption sites investigated?
- 4) do the authors include zero point energy (ZPE) and entropic contributions in the calculation of the Gibbs free energy?
- 5) what is the effect of the electrolyte on the reaction energetics? A more realistic surface (with electrolyte) should be studied.

Reviewer #2 (Remarks to the Author):

Comments:

The authors have properly addressed all my concerns, this reviewer thus recommended its acceptance on nature communications.

Response: Thank you very much for the positive comment and kind recommendation.

Reviewer #4 (Remarks to the Author):

Comments:

In the manuscript "Active site recovery and N-N bond breakage during hydrazine oxidation boosting the electrochemical hydrogen production", Zhu and co-workers have investigated the use of a new nanostructured interface as alternative catalysis to boost the HER/HzOR electrocatalysis. The approach is clearly interesting and the topic is very timely as green-hydrogen production is high in nowadays challenges. The manuscript is well-written, however, is rather lengthy and the conclusions are diluted in the text. The computational part, for example, is using a rather simple model and the computational conclusions are rather detached from the rest of the text. I would recommend the authors integrate better the different aspects of the study before publication. Moreover, the following points should be addressed:

Response: Thank you very much for the positive comment and kind recommendation.

We have revised the section of "**H₂OR mechanistic insight in high stability and activity**". Based on the revisions, the conclusion of this work become more convincing:

(1) the bimetallic phosphide catalysts (Ni-Co-P/NF) with heterogeneous interfaces have been prepared to achieve excellent HzOR/HER activity. (2) The mechanism of the attractive HzOR activity of the catalyst has been explored in depth: The durability is derived from the continuous re-exposure of active center (metal phosphide); The excellent activity comes from the modulated electronic structure at the heterointerface accelerating the dehydrogenation kinetics in the traditional path, and an alternative N-N bond breakage pathway has been proposed under the anodic voltage. (3) Based on

the excellent performance, an efficient hydrogen generation system without external power supply is built.

In the computation, we have mainly studied the catalytically active site preference of HzOR on the constructed heterostructure based on the traditional HzOR reaction path, and verified the existence of the N-N bond breakage path on heterointerface. In the revised manuscript, the calculation model and the relevant results have been optimized in **line 389-426, Page 17-18**. (The optimized models have been corrected in **Fig. 5f, Fig. 5g, Fig. S25, Fig. S26, Fig. S27, Fig. S29, Fig. S30, Fig. S31, Fig. S32**.) The detailed descriptions about the calculation model and DFT methods have been supplemented in **line 653-697, Page 27-28** in the revised manuscript. Also, the effect of electrolyte on the calculation result has been fully considered, making the work more integrated.

We hope the revision have addressed your valuable comments. Please find our point-by-point responses below.

1. how did the authors obtain the interface used as a computational model? It is not clear how realistic and representative the model is.

Response: Thank you very much for your question and reminding. The interface used as the initial computational model was obtained from the XRD spectra in **Fig.2a**, in which the (011) plane of CoP and (111) plane of NiCoP are lattice planes of the strongest peak intensities, showing that they are the most likely exposed facets of the catalyst offering adsorption sites in the heterostructure model. Moreover, the strong interfacial contact between NiCoP (111) and CoP (011) facets could be found in the high-resolution TEM images in **Fig. S8**. Therefore, the NiCoP(111)/CoP(011) interface was selected in this study to simulate the NiCoP-CoP heterostructure of the Ni-Co-P/NF catalyst.

In considering the influence of electrolyte solution, we have finely tuned the original model to make the model more representative, as shown in **Fig. S25**. Relevant discussion has been supplemented in **line 391-393, Page 17** in the revised Manuscript. In addition, the model with NiCoP(111)/CoP(011) interface has also been reported in

the NiCoP/CoP heterojunction system (*Electrochim. Acta*, 2021, 390, 138840, *Chem. Eng. J.*, 2022, 446, 137294).

Fig. S8. HRTEM image of Ni-Co-P/NF with interface division.

Fig. S25. The structural model to simulate the Ni-Co-P/NF. (a) Initial model; (b) optimized model.

2. how stable is the structure when in contact with the electrolyte? Is there leakage of Ni or Co?

Response: Thank you very much for the questions. We have applied the Ab initio molecular dynamics (AIMD) calculations to simulate the structure stability in contact with electrolyte, with a layer of solution (including 15 water molecules and one OH group) covered on the NiCoP(111)/CoP(011) surface (**Fig. S26 a, b, c**). By 20000 steps of kinetic relaxation, the solution molecules are dispersed in the upper vacuum, while the NiCoP-CoP model still maintains integrated with a slight change of Gibbs free energy ($\Delta G = -0.029$ eV) after kinetic relaxation. AIMD calculation result confirms the

high stability of the heterojunction structure in alkaline aqueous solution. Relevant discussion has been supplemented in **line 393-395, Page 17** in the revised Manuscript.

On the other hand, the as-synthesized catalyst has been confirmed to be stable in the long terms of *i-t* measurements for HzOR as shown in **Fig. S18, Fig. S19, Fig. S20**. And the time-dependent concentrations of Ni and Co in the electrolyte solution of Ni-Co-P/NF electrode has been supplied in **Fig. S26d**, showing that under the influence of electrolyte, negligible amounts of Ni and Co dissolutions can be found from the catalyst. Therefore, both the catalyst composition and the structure in the calculation model are highly stable in the electrolyte solution.

Fig. S26. (a) Ab initio molecular dynamic (AIMD) models to simulate the structure in contact with electrolyte, with a layer of solution (including 15 water molecules and one OH groups) covered on the NiCoP(111)/CoP(011) surface; (b) Model by 20000 steps of kinetic relaxation; (c) AIMD calculation results. (The total energy has a small drift in AIMD calculations, which is caused by the diffusion of water molecules. The overall energy remains highly stable.) (d) Curves of time-dependent concentration of Ni and Co in the electrolyte solution of Ni-Co-P/NF electrode during the *i-t* test for HzOR.

3. how representative are the adsorption sites investigated?

Response: Thank you very much for the question. From the Bader charge analysis (**Fig. S28**), more charges from Co can transfer to P than from Ni, showing that the N_2H_4 will be preferentially adsorbed on the Co site in the heterostructure, followed by the Ni site. This phenomenon is consistent with the reports in literatures that the adsorption site of HzOR in cobalt-nickel catalyst system is located on Co sites (*Angew. Chem. Int. Ed.*, 2021, 60 (11), 5984-5993. *Nat. Commun.*, 2020, 11 (1), 1853. *Chinese J. Catal.*, 2022, 43 (4), 1131-1138).

Furthermore, in the calculation model, there are five different kinds of sites at the NiCoP(111)/CoP(011) interface as shown in **Fig. S29**. We have calculated the adsorption energies of N_2H_4 on these sites, and found that $\Delta G^*_{N_2H_4}$ at the Co site (-0.530 eV) is optimal, confirming the representative adsorption site of Co toward HzOR. Relevant discussion has been supplemented in **line 398-405, Page 17** in the revised Manuscript.

Fig. S29. The five different sites at the NiCoP(111)/CoP(011) interface (a, b) and the change of free-energy for N_2H_4 adsorption on the sites (c). The $\Delta G^*_{N_2H_4}$ on P1, P2 and P3 sites means that the P atom-terminated unit cell is not favorable for N_2H_4 adsorption, which is inconsistent with experimental results.

4. do the authors include zero point energy (ZPE) and entropic contributions in the calculation of the Gibbs free energy?

Response: Thank you very much for your professional question. We have considered the impact of ZPE and entropic contributions in the calculation of the Gibbs free energy. The values of ΔS for gas phase H_2 , N_2 and N_2H_4 were obtained from the NIST-JANAF thermodynamics table, and ΔZPE was calculated by the vibration contribution of all adsorbed species. The relevant description has been supplemented in **line 683-687, Page 28** in the revised Manuscript.

5. what is the effect of the electrolyte on the reaction energetics? A more realistic surface (with electrolyte) should be studied.

Response: Thank you very much for the constructive question. The effect of the electrolyte on the reaction energetics has been investigated and added in the revised Manuscript (**line 689-693, Page 28**) and Supporting Information (**Page 20**). The effects of electrolyte on the HzOR can be considered in two aspects: cations effects (K^+) and pH effects (OH^-). (*Acc. Chem. Res.*, 2022, 55 (4), 495-503)

As for the effect of cations (K^+) on the kinetics in alkaline media, we tuned the pH of the alkaline solution to 12, and then increases the concentration of K^+ in the electrolyte and conduct the electrochemical tests (*Angew. Chem. Int. Ed.*, 2021, 60 (24), 13452-13462). The results are shown in **Fig. S33**. The K^+ mainly affects the ionic conductivity of the solution (the solution resistance R_s), not the charge-transfer resistance (R_{ct}) according to the result of EIS and model fitting (**Fig. S33b**). Thus, the Tafel plots after iR_s compensation in electrolyte are almost the same under different K^+ concentrations (**Fig. S33d**). The result shows that the influence of K^+ on the reaction kinetics is rather weak, which can be ignored when calculating the energy.

For a multistep proton–electron-transfer reaction, the reaction thermodynamics depend on the potential vs the reversible hydrogen electrode (RHE), and is therefore affected by both the proton activity (pH) of the electrolyte and the absolute potential of the electrode (*Acc. Chem. Res.*, 2022, 55 (4), 495-503). In this work, the influence of pH has been fully considered when calculating the energy change, and relevant

descriptions have been supplemented in the revised Manuscript (line 689-691, Page 28).

In addition, the anions and cations in the electrolyte may form a double-layer capacitance, which will affect the reaction energetics, and this effect is represented by ΔG_{field} in calculations (line 691-693, Page 28) (*ACS Catal.*, 2016, 6 (10), 7133-7139). Therefore, in the calculation, we have supplemented an additional electric field (-0.2 V/\AA) in the adsorption process of N_2H_4 molecule to simulate the possible double-layer capacitance, and the adsorption energy change of N_2H_4 has been calculated to be as low as 0.051 eV. The result shows the negligible effect of electric field on the adsorption of hydrazine molecules.

According to the above experiment and theory calculation, the cations in the electrolyte and the potential formation of the double-layer capacitance have negligible influences on the reaction energy, and therefore ignored during the calculation. The effect of anion (pH) on the reaction thermodynamics has been considered in the Gibbs free energy calculation.

Fig. S33. Electrochemical tests for HzOR on Ni-Co-P/NF surface in 10 mM KOH (pH=12), 10 mM KOH + 90 mM KClO₄ (pH=12), 10 mM KOH + 990 mM KClO₄ (pH=12). (a) LSV curves without iR compensation; (b) Nyquist plots; (c) LSV curves with iR compensation; (d) Tafel plots after iR compensation.

Thanks again for your time and effort. Following your suggestions, we have supplemented major functional and mechanistic studies as presented above. We hope that the revision could well-address your concern.

REVIEWERS' COMMENTS

Reviewer #4 (Remarks to the Author):

All the comments of the reviewers have been addressed in a satisfactory way. I recommend the paper for publication.

Reviewer #4 (Remarks to the Author):

Comments:

All the comments of the reviewers have been addressed in a satisfactory way. I recommend the paper for publication.

Response: We thank the reviewer for reviewing and recommending to our work.